# Inter–annual global carbon cycle variations linked to atmospheric circulation variability

Na Li[1], Sebastian Sippel[2], Alexander J. Winkler[1], Miguel D. Mahecha[3], Markus Reichstein[1], and Ana Bastos[1]

[1]Max Planck Institute for Biogeochemistry, Germany
[2]Institute for Atmospheric and Climate Science, and Seminar for Statistics, ETH Zurich, Switzerland
[3]Remote Sensing Center for Earth System Research, Leipzig University, Germany

**Correspondence:** Na Li (nali@bgc-jena.mpg.de)

**Abstract.** One of the least understood temporal scales of global carbon cycle (C–cycle) dynamics is its inter–annual variability (IAV). This variability is mainly driven by variations in the local climatic drivers of terrestrial ecosystem activity, which in turn are controlled by large–scale modes of atmospheric variability. Here, we quantify the fraction of global C–cycle IAV that is explained by large–scale atmospheric circulation variability, which is quantified by spatio–temporal sea level pressure (SLP) fields. C-cycle variability is diagnosed from the global detrended atmospheric $CO_2$ growth rate and the land $CO_2$ sink from 16 dynamic global vegetation models and two atmospheric inversions in the Global Carbon Budget 2018. We use a regularized linear regression model, a statistical learning technique apt to deal with the large number of atmospheric circulation predictors ($p \geq 800$, each representing one pixel–based time–series of SLP anomalies) in a relatively short observed record ($n < 60$ years). We show that boreal winter and spring SLP anomalies allow predicting IAV in atmospheric $CO_2$ growth rate and of the global land sink, with Pearson correlations between reference and predicted values between 0.70–0.84 boreal winter SLP anomalies. This is comparable or higher than that of a similar model using 15 traditional teleconnection indices as predictors. The spatial patterns of regression coefficients of the model based on SLP fields show a predominant role of the tropical Pacific and over Southeast Asia extending to Australia, corresponding to the regions associated with the El Niño/Southern Oscillation variability. We also identify another important region in the western Pacific, roughly corresponding to the West Pacific pattern.

We further evaluate the influence of the time–series length on the predictability of IAV and find that reliable estimates of global C–cycle IAV can be obtained from records of 30–54 years. For shorter time–series ($n < 30$ years), however, our results show that conclusions about $CO_2$ IAV patterns and drivers need to be evaluated with caution. Overall, our study illustrates a new data–driven and flexible approach to model the relationship between large–scale atmospheric circulation variations and C–cycle variability at global and regional scales, complementing the traditional use of teleconnection indices.

## 1 Introduction

The global carbon cycle (C–cycle) varies at multiple time scales ranging from minutes to millennium (Ciais et al., 2013). Quantifying and understanding the patterns of variability in the C–cycle and their drivers is crucial to better understand C–cycle dynamics and better constrain future climate projections (Cox et al., 2013; Friedlingstein et al., 2014). Primarily driven by the

land sink dynamics (Piao et al., 2020), inter–annual variability (IAV) is one of the most uncertain and poorly understood terms of the global C–cycle in the observational period (Friedlingstein et al., 2019).

A fundamental challenge is that variability in land–atmosphere carbon exchange is affected in complex ways by large–scale atmosphere circulation modes but also land use change, forced climate changes, direct physiological $CO_2$ effects on ecosystems, among others (IPCC, 2013). Separating these effects is difficult because of the large uncertainties associated with some processes, for example land use change (Friedlingstein et al., 2022) or processes not directly observable at global scale (e.g., photosynthesis or heterotrophic respiration) (Schimel et al., 2015; Basile et al., 2020). The second challenge is that the land sink, as a balance of carbon uptake and release, responds differently to variations in the climatic forcing (Jung et al., 2017; Piao et al., 2020). This makes it hard to attribute induced land sink IAV to specific drivers, which is crucial for process understanding (Jung et al., 2017; Humphrey et al., 2018, 2021; Wang et al., 2022). Last, the limited length of observational records may hamper robust statistical analysis (IPCC, 2013): the longest continuous observations of atmospheric $CO_2$ at the South Pole and Mauna Loa observatory exist only since 1958 onwards (Dlugokencky and Tans, 2019).

Land biospheric $CO_2$ uptake results from the net balance of carbon uptake from gross primary productivity, and release from multiple respiration terms, and disturbance induced fluxes such as fires, amongst other smaller terms (IPCC, 2013). Most of these processes are primarily driven by temperature, water and radiation availability (Jung et al., 2017). These meteorological drivers are, in turn, modulated by large–scale modes of atmospheric circulation on multiple time–scales, such as El Niño/Southern Oscillation (ENSO; Gu and Adler, 2011) and the Pacific Decadal Oscillation (PDO; Newman et al., 2016). These climate variability modes are generated within the coupled atmosphere–ocean systems (Ghil, 2002) and considered irreducible noise in climate projections (Madden, 1976; Schneider and Kinter, 1994; Deser et al., 2012). Because these modes typically interact and affect weather dynamics in regions beyond those where they are emerged, such modes are collectively referred to as teleconnections (IPCC, 2013). Bacastow (1976) showed that ENSO is highly correlated with annual variations in observed atmospheric $CO_2$ measured at the South Pole and Mauna Loa, Hawaii. Keeling et al. (1995) attributed these correlations to the ENSO impact on the biospheric sink. In addition to ENSO, Zhu et al. (2017) showed that the PDO and the Atlantic Multi–decadal Oscillation (AMO; Enfield et al., 2001; Rayner et al., 2003) may also influence global terrestrial ecosystem carbon fluxes and that other modes of variability in the Northern Hemisphere have also local impacts on carbon cycling (Zhu et al., 2017).

A common approach to diagnose the impacts of natural climate variability is to use ensembles of Earth system model simulations with perturbed initial conditions to quantify the impacts of natural climate variability at decadal to millennium scales (Frölicher et al., 2013). However, the inherently chaotic atmosphere, in combination with model structural uncertainty, implies large uncertainties for future projections (Deser et al., 2020). In addition, Earth system model projections can be compromised by limited representation of the full complexity of physical processes involved, lack of observational constraints, and high computational demands when aiming to resolve high resolutions (Randall et al., 2007; IPCC, 2013).

Statistical approaches are a simplified but effective way to reveal physical processes in observations (von Storch, 1995). A traditional approach consists of evaluating relationships between the variables of interest (e.g. $CO_2$ time–series) and teleconnection indices (Bacastow, 1976; Bastos et al., 2013; Zhu et al., 2017). As a simple representation of the large–scale atmo-

spheric circulation modes, teleconnection indices are extracted mainly from sea surface temperature or atmospheric anomalies (Kumar and Hoerling, 1997; IPCC, 2013). Such indices are an effective way to reduce the complexity of the spatio–temporal variability in multiple variables (Stenseth et al., 2003; Wills et al., 2017), but may not be able to capture spatial variations in the large–scale atmospheric circulation modes themselves.

Recently, Sippel et al. (2019) applied Ridge Regression, a regularized linear regression method (Hastie et al., 2009; Friedman et al., 2010), to quantify the component of precipitation and temperature variability driven by atmospheric variations based on sea level pressure (SLP) fields, rather than teleconnection indices. Their approach allowed them to robustly infer the main spatio–temporal patterns of atmospheric variability influencing these two climate variables. On the one hand, including a field of circulation–based predictors, avoids considering predefined assumptions about their spatial configurations as they are common to teleconnection indices, while compensating for relatively short historical records. The regularization approach, on the other hand, allows to overcome overfitting and multicollinearity issues due to short time–series and a very large number of spatial predictors.

In this study, we adopt the Ridge Regression approach in Sippel et al. (2019), aiming to quantify the fraction of global C–cycle IAV influenced by large–scale atmospheric circulation variability. We use observation–based time–series of global atmospheric $CO_2$ growth rate ($AGR$) and land $CO_2$ surface fluxes from atmospheric inversions and Dynamic Global Vegetation (DGVMs), as well as the land sink estimated as a residual of other terms in the Global Carbon Budget 2018 (Le Quéré et al., 2018). We first evaluate and compare the predictive skill of the Ridge Regression model when using SLP as predictors versus commonly used traditional teleconnection indices (Section 3.1). Next, we analyze and discuss the global C–cycle sensitivity to atmospheric circulation variability from various latitudinal domains of SLP anomaly fields (Section 3.2). Finally, we evaluate the sensitivity of the results to the length of the time–series (Section 3.3), by comparing the fraction of C–cycle IAV that can be explained by large–scale atmospheric circulation variability based on these datasets with that of a very long time–series (4000 years) of land $CO_2$ fluxes simulated by the Community Earth System Model (CESM).

## 2 Data and methods

### 2.1 $CO_2$ datasets for the recent past

We select the $CO_2$ time–series datasets from the Global Carbon Budget 2018 version 1.0 (Le Quéré et al., 2018): the atmospheric $CO_2$ growth rate ($AGR$), the land sink from models ($SL_{\text{DGVMs}}$), the residual land sink ($SL_{\text{Resid}}$), and the land sink from two atmospheric inversions.

In the Global Carbon Budget 2018 (Le Quéré et al., 2018), the global $CO_2$ balance is calculated based on the carbon emissions from fossil fuel ($FF$) (Boden et al., 2017; UNFCCC, 2018; Peters et al., 2011b) and land use change ($FLUC$) (Houghton and Nassikas, 2017; Hansis et al., 2015), the $AGR$ (Dlugokencky and Tans, 2018), the carbon uptake by the ocean sink ($SO$) and the land sink ($SL$) (references for individual models of $SO$ and $SL$ can be found in Table 4 of Le Quéré et al. (2018)).

The difference of annual atmospheric $CO_2$ in a given year and the previous year (Ballantyne et al., 2012; Dlugokencky and Tans, 2018; Le Quéré et al., 2018) corresponds to the $AGR$, which is based on direct observations. The $AGR$ is based on the average of well–mixed $CO_2$ measurements at multiple global stations from the US National Oceanic and Atmospheric Administration Earth System Research Laboratory (NOAA ESRL) (Dlugokencky and Tans, 2018).

$FF$ emissions are based on inventories, while $FLUC$, $SL$ and $SO$ are estimated by models ($SL_{\mathrm{DGVMs}}$ and $SO$, respectively in Eq. (1)), all of which contain uncertainties (Le Quéré et al., 2018). The total emissions from $FF$ and $FLUC$ minus $AGR$ should equal the $SO$ and $SL_{\mathrm{DGVMs}}$ (Eq. (1)). Due to uncertainties in modeled land and/or ocean sinks or in land use estimations (Bastos et al., 2020; Hauck et al., 2020), the budget cannot be balanced and thus an imbalance term ($IMB$) is introduced to the budget.

$$FF + FLUC - AGR - SO = SL_{\mathrm{DGVMs}} + IMB = SL_{\mathrm{Resid}} \tag{1}$$

The annual land sink of $CO_2$ ($SL_{\mathrm{DGVMs}}$) is the average net biome production (NBP) simulated by 16 dynamic global vegetation models (DGVMs) forced with historical $CO_2$ and changing climate (Le Quéré et al., 2018). The residual land $CO_2$ ($SL_{\mathrm{Resid}}$) is calculated from emissions, $AGR$, and ocean sinks, as described in Eq. (1). $SL_{\mathrm{Resid}}$ corresponds to the balance of the fossil fuel and land–use change emissions and the sinks in the atmosphere and ocean and provides an alternative estimate of the global land sink.

The time–series of $AGR_{\mathrm{R}}$, $SL_{\mathrm{DGVMs}}$, and $SL_{\mathrm{Resid}}$ in Global Carbon Budget 2018 (Le Quéré et al., 2018) are provided at annual time–steps over the period 1959–2017. In the following analysis, we invert the $AGR$ time–series ($AGR_{\mathrm{R}}$ for reversed $AGR$ i.e. -1×$AGR$) for sign consistency with the land sink datasets used (defined as a positive flux from the atmosphere to the land).

Additionally, we use the globally aggregated net atmosphere to land $CO_2$ flux (positive sign as a sink in the biosphere) estimated from two atmospheric $CO_2$ inversions in Global Carbon Budget 2018 (Le Quéré et al., 2018): the Jena CarboScope $SL_{\mathrm{CarboScope}}$ (Rödenbeck, 2005; Rödenbeck et al., 2018), and the Copernicus Atmosphere Monitoring Service inversion $SL_{\mathrm{CAMS}}$ (Chevallier et al., 2005), which cover the periods 1976–2017 and 1979–2017 respectively. Here we use the global annual $CO_2$ fluxes of these two inversions adjusted for fossil fuel emissions and lateral fluxes from Bastos et al. (2020). The period common to the $CO_2$ time–series (1980–2017) is selected.

### 2.1.1 Sea level pressure

We use global monthly mean SLP fields from ERA5 reanalysis with the spatial resolution of 0.25°×0.25° (Bell et al., 2020), at monthly time–steps and covering the period 1950–1978 (Bell et al., 2020) and 1979–present (Hersbach et al., 2019). The period common to other datasets of 1958–2017 is selected here.

### 2.1.2 Teleconnection indices

In addition to SLP fields, we select 15 teleconnection indices from the atmosphere–ocean variability, Northern Hemisphere, and Southern Hemisphere.

Three important atmosphere–ocean coupled variability modes influence global climate and the C–cycle: the El Niño–Southern Oscillation (SOI), the Pacific Decadal Oscillation (PDO), and the Atlantic Multidecadal Oscillation (AMO) (Zhu et al., 2017).

In the Northern Hemisphere, the most relevant indices are: the Arctic Oscillation (AO), the North Atlantic Oscillation (NAO), the Pacific North American pattern (PNA), the East Atlantic (EA), the East Atlantic/Eastern Russia (EAWR), the Scandinavian pattern (SCAND), the Polar/Eurasia (polarEA), and the West Pacific (WP). These indices are calculated and provided by the Climate Prediction Centre (CPC) of the National Oceanic and Atmospheric Administration (NOAA)(CPC, 2008). The detailed information on calculation procedures is described on NOAA CPC (2008) and Barnston and Livezey (1987).

In the Southern Hemisphere, important indices are the Antarctic Oscillation (AAO), the Tropical Atlantic Dipole (TAD), the Dipole Mode Index (DMI) of the Indian Ocean Dipole, and the Trans Polar index (TPI).

The teleconnection indices used here have been summarized in Table 1. All the indices are provided as monthly means and selected for the period of 1958-2017, except the AAO which is available for 1979–2017 only.

### 2.1.3 Long–term pre–industrial control simulations for statistical benchmarking

Here we select the SLP fields and global net biome production fields (NBP) from simulations by the Community Earth System Model (CESM) version 1.2.2 (in the B1850C5CN configuration), which has been used by Stolpe et al. (2019). This experiment corresponds to a 4000-yr control run. The simulation was run at an atmospheric resolution of $1.9° \times 2.5°$, using the Community Atmosphere Model version 5 (CAM5.3; (Neale et al., 2012)) with 30 vertical levels. The model consists of fully coupled atmosphere, ocean, sea ice and land surface components (Hurrell et al., 2013; Meehl et al., 2013b), and did not include dynamic vegetation. This simulation includes no external forcing, so it is ideal to analyze patterns driven by internal variability.

## 2.2 Data pre–treatment

For all historical datasets ($CO_2$ time–series, SLP fields and teleconnection indices), we first remove years corresponding to volcanic eruptions (1963, 1982, 1983, 1991, 1992). We then pre–treat the datasets as follows.

### 2.2.1 Trend removal

The long–term trend of $CO_2$ time–series is removed by locally weighted scatterplot smoothing (LOWESS) (Cleveland et al., 1991) of the annual time–series with fixed window size of 25 % interval longer than 30 years (1959–2017) and 45 % for shorter period (1980–2017). For monthly teleconnection indices, we first calculate the seasonal mean values of DJF, MAM, JJA, and SON, we then remove the seasonal long–term trends by applying the LOWESS as for the $CO_2$ time–series, and further include DJF and MAM combined (DJF+MAM) as treated in SLP (as described below).

### 2.2.2 Spatial and temporal aggregation

The monthly mean SLP fields are area–weighted and aggregated to $2° \times 2°$, $5° \times 5°$, and $9° \times 9°$ spatial resolution, and the seasonal cycle removed by subtracting the monthly mean values for each pixel. We then aggregate SLP values in seasonal means for: December of the previous year to February of each given year (DJF), March–May (MAM), June–August (JJA), and September–November (SON) and further consider DJF and MAM combined (DJF+MAM), so the number of pixel–based time–series (predictors) in DJF+MAM is double of DJF. Note that a large fraction of the pixel–based time–series of seasonal SLP anomalies show no long–term trend, and the predicted differences between LOWESS detrended and not detrended SLP are small. Here we keep the analysis of SLP anomalies with no LOWESS detrending. Here, we refer to DJF and MAM as boreal winter and boreal spring.

For the CESM simulations, the SLP fields are originally provided at $1.9° \times 2.5°$ spatial resolution at monthly mean time–steps, which we then resample to $5° \times 5°$ spatial resolution. Annual mean NBP is calculated from the monthly fields. NBP and SLP fields are selected for the simulation period 1000–5000 year.

### 2.3 Statistical analysis

The overall goal is to characterize annual variations in the global C–cycle that can be explained by large–scale atmospheric circulation variability. Here, the pixel–based time–series of SLP anomalies are used as predictors ($p \geq 800$) of $CO_2$ time–series ($n \leq 54$ years) in a linear regression model. However, the small sample size relative to the large number of predictors ($n < p$) can cause severe overfitting problems and result in unstable predictions (Hastie et al., 2009). Moreover, the existing spatial correlations among the neighboring pixels of SLP anomalies might cause multicollinearity among the predictors (von Storch and Zwiers, 1999). The potential multicollinearity problem results in unstable Ridge Regression coefficients in least square estimation, and making it difficult to diagnose the most sensitive spatial patterns of predictors (von Storch and Zwiers, 1999).

Sippel et al. (2019) applied Ridge Regression to avoid these overfitting and multicollinearity problems. Ridge Regression is a regularized linear regression, whose the fundamental principle is to introduce a constraint (hyper–parameter $\lambda$) to regularize the varying regression coefficients in least squares estimation (Hastie et al., 2009; Friedman et al., 2010). The regularized variance comes with a compromise of biased predictions and is addressed as the bias–variance trade–off (Hastie et al., 2009). When selecting the best hyper–parameter $\lambda$, this trade–off is considered to achieve stable (low variance) while slightly biased predictions (Hastie et al., 2009).

Model performance is evaluated by the $R^2$, the Pearson's correlation $R$, and mean squared error of the original $CO_2$ time–series against predicted values. Pearson's correlation $R$ is selected as the main measure of predictability, and the significance $P < 0.05$ is selected. Given the relatively short period ($n < 60$), here we use leave–one–out cross–validation to achieve optimal model training and testing. For each train and test group splitting, we select the train group as all years excluding three consecutive years and the middle year of those three is then selected as the test sample. We exclude the preceding and following years to reduce the potential influence of temporal auto–correlation.

**Table 1.** Teleconnection indices

| Index | Name | Description | Source |
|-------|------|-------------|--------|
| SOI | Southern Oscillation | Monthly sea level pressure anomalies differences (based on 1981–2010 monthly mean) between Tahiti and Darwin, Australia. (McBride and Nicholls, 1983; Ropelewski and Jones, 1987) Downloaded from the NOAA National centers for Environmental Information (NCEI). | https://www.ncdc.noaa.gov/ teleconnections/enso/indicators/ soi/#soi-calculation |
| PDO | Pacific Decadal Oscillation | Monthly sea surface temperature (SST) variations in the Northeast and tropical pacific Ocean (Mantua et al., 1997; Mantua and Hare, 2002). Using EOF and regression over $20°$–$90°$ N in the Pacific (Mantua et al., 1997). Downloaded from NOAA NCEI. | https://www.ncdc.noaa.gov/ teleconnections/pdo/ |
| AMO | Atlantic multi-decadal Oscillation | Monthly Northern Atlantic temperature fluctuations (Rayner et al., 2003; Enfield et al., 2001). Computed by NOAA Physical Science Laboratory (PSL) (using Kaplan SST V2 dataset) from $0°$–$70°$ N (Enfield et al., 2001). The detrended and unsmoothed version is selected. | https://psl.noaa.gov/gcos_wgsp/ Timeseries/AMO/ |
| AO | Arctic Oscillation | Characterized by winds circulations near the Arctic around $55°$ N. Calculated by NOAA NCEI, using Empirical Orthogonal Function (EOF) analyzes the monthly mean 1000 millibar height variations over $20°$–$90°$ N (Higgins et al., 2000, 2002). | https://www.ncdc.noaa.gov/ teleconnections/ao/ |
| NAO | North Atlantic Oscillation | The Subtropical High and the Subpolar Low difference in sea level pressure (Barnston and Livezey, 1987). Downloaded from NOAA NCEI. | https://www.ncdc.noaa.gov/ teleconnections/nao/ |
| PNA | Pacific-North America | Low–frequency Variations in the Northern Hemisphere extratropics (Barnston and Livezey, 1987; Chen and Van den Dool, 2003). Downloaded from NOAA NCEI. | https://www.ncdc.noaa.gov/ teleconnections/pna/ |
| EA | East Atlantic | North–south dipole anomalies extending from the east to west North Atlantic, with a similar spatial structure to NAO (Barnston and Livezey, 1987). | https://www.cpc.ncep.noaa.gov/ data/teledoc/ea.shtml |
| EAWR | East Atlantic/Western Russia | Distinct by four dominant anomaly centers with positive phase extending Europe and Northern China, negative covering central North Atlantic and North Capspian Sea (Barnston and Livezey, 1987). | https://www.cpc.ncep.noaa.gov/ data/teledoc/eawruss.shtml |
| SCAND | Scandinavia | A main anomaly center over Scandinavia, and a opposite weaker sign over western Europe and eastern Russia/western Mongolia (Barnston and Livezey, 1987). | https://www.cpc.ncep.noaa.gov/ data/teledoc/scand.shtml |
| PolarEA | Polar/Eurasia | In positive pattern, negative height anomalies in polar region, and positive anomalies in Northern China and Mongolia (Barnston and Livezey, 1987). | https://www.cpc.ncep.noaa.gov/ data/teledoc/poleur.shtml |
| WP | West Pacific | Low frequency variability of North Pacific (Barnston and Livezey, 1987). | https://www.cpc.ncep.noaa.gov/ data/teledoc/wp.shtml |
| AAO | Antarctic Oscillation | Empirical Orthogonal Function (EOF) was applied to the monthly mean 700hPa height anomalies over $20°$–$90°$ S (Mo, 2000). Calculated by NOAA (CPC, 2008). | https://www.cpc.ncep.noaa.gov/ products/precip/CWlink/daily_ao_ index/aao/aao.shtml |
| TAD | Tropical Atlantic Dipole | Here we use TSA (Tropical Southern Atlantic Index). Obtained from NOAA Physical Sciences Laboratory (PSL), computed from the monthly SST average anomaly from $0°$–$20°$ S and 10 $°$ E–$30°$ W, HadISST and NOAA OI 1x1 datasets are used (Enfield et al., 1999; Reboita et al., 2021). | https://psl.noaa.gov/data/ correlation/tsa.data |
| DMI | Dipole Mode | Obtained from NOAA PSL, based on the sea surface temperature anomaly gradient between the western and the South eastern equatorial Indian Ocean, HadISST1.1 SST is used (Saji and Yamagata, 2003; Reboita et al., 2021). | https://psl.noaa.gov/gcos_wgsp/ Timeseries/Data/dmi.had.long. data |
| TPI | Trans Polar | Normalized pressure difference between stations in Hobart and Stanley, Australia (Pittock, 1980, 1984; Jones et al., 1999), data calculated by Henley et al. (2015), data from University of East Anglia http://www.cru.uea.ac.uk/cru/data/tpi/. | https://psl.noaa.gov/gcos_wgsp/ Timeseries/TPI/ |

A schematic description of the workflow from model training, validation and selection with the selected interval 1959–2017 is shown in Fig. 1. In panel (a), the global maps represent an example of the spatial distribution of SLP with a resolution of $m° \times m°$, where $m$ varies in [2, 5, 9]. Each pixel corresponds to a time–series of SLP, so that $p$ predictors ($x_{ik}$, for $i$ in 1~n

185    years, and $k$ in 1~p, where p depends on the spatial resolution selected) with $n = 54$ time–steps are defined. Each predictor is assigned a coefficient $\omega_k$ to collectively predict $CO_2$ time–series, with length $n = 54$. The cost function (Harrington, 2012) is the sum of all the squared errors of original $y_i$ minus estimated $y_i'$ ($x_i^T \omega + b$, $\omega$ represents the vector of all $\omega_k$). At the same time, the constraint function (Harrington, 2012) suppresses the coefficient variations under a regularized range defined by the hyper–parameter $\lambda$. Panel (b) shows the model training and validation processes. In this study, SLP is aggregated to $9° \times 9°$, so

$p = 800$. Note that to reduce the heavy computation load, when conducting spatial and temporal sensitivity study as described in Section 2.4, the range of $\lambda$ is lower and with smaller steps than the range and step shown in the Panel (b). For example, when only selecting the tropical domain of SLP, the number of predictors is much less than 800. So the $\lambda$ is selected in range [10, 1000] with a step of 50. When using teleconnection indices instead of SLP anomalies, the predictors are equal or less than 15, the range of $\lambda$ is selected from [1, 200] with a step of 2.

The first step is to divide datasets into train and test groups, as shown in panel (c). The grouped training datasets are then used for model training and tune the best $\lambda$ through 5–fold cross–validation, the best $\lambda$ that achieves the optimal prediction (highest $R^2$) is then selected by the model to predict test datasets. The model then starts another iteration with train and test grouping. Panel (c) describes the leave–one–out train and test grouping, the years before and after the selected test year are removed for each grouping to reduce the impact of temporal auto-correlations in $CO_2$ time–series.

Ridge Regression leave–one–out cross–validation is performed using the Python package Scikit–learn "Ridge" and the $\lambda$ is tuned by Scikit–learn "RidgeCV" (Pedregosa et al., 2011). The global maps (Fig. 3, Fig. A4, Fig. A11, Fig. A12) are plotted by Cartopy (Met Office, 2010–2015).

## 2.4   Experimental design

    1. Preliminary dependency tests. To evaluate the robustness of the results for different characteristics of the datasets and

methodological choices, we perform several preliminary tests. **1)** Resolution–dependency test: evaluate the sensitivity of results to the SLP spatial resolution under $2° \times 2°$, $5° \times 5°$, and $9° \times 9°$; **2)** Seasonality–dependency test: evaluate the dependence of results on the definition of particular seasons, with each season being the combination of three consecutive months (from November last year to July the given year). **3)** Temporal auto–correlation of the $CO_2$ time–series: to ensure no significant trend remains in the detrended $CO_2$ time–series.

Here we directly use the results following the preliminary dependency test (Appendix A). The spatial resolution does not influence the results considerably (see Appendix A Fig. A1), therefore we select $9 \times 9°$ SLP spatial resolution given its smaller number of grid points. The seasonal dependency test shows that DJF and MAM are seasonal combinations more representative of boreal winter and spring (see Appendix A Fig. A2). JJA and SON are found to have lower or no predictability to $CO_2$ time–series, therefore, we limit our results to DJF and MAM. As shown in Appendix A Fig. A3, the

temporal auto–correlation of all $CO_2$ time–series is mostly less than 0.4 with lag ranging from 1 to 35 years. With a lag of one year, absolute values of auto–correlation are below 0.2, so that we can exclude strong temporal auto–correlation effects.

2. Model training and evaluation. We evaluate the predictability of annual $CO_2$ time–series using SLP anomalies, teleconnection indices, and SOI independently in the DJF, MAM, and DJF+MAM seasons, using the approach described above. We compare Pearson's correlation of observations and predicted values for different periods (1959–2017 and 1980–2017, Fig. 2 b) and show the corresponding Ridge Regression coefficient distribution maps (Fig. 3 a).

3. Spatial sensitivity study. We evaluate the predictability of historical annual $CO_2$ time–series using DJF SLP anomalies under different spatial domains in periods of 1959–2017 and 1980–2017. Then we use a 30–yr sliding window with annual $AGR_R$ to depict how the predictability under various SLP domains evolves in the period 1959–2017.

4. Temporal sensitivity study. We evaluate the predictability of annual $CO_2$ time–series $AGR_R$, $SL_{DGVMs}$, and CESM using DJF and MAM SLP anomalies under different time intervals. Sliding windows are employed at time intervals of 15, 20, 30, 40 years for historical datasets and CESM, and 100, 500, and 2000 years for CESM only. For the interval of 100 and 500 years, we use the sliding window of a 50 year step, and a 500 year step for the 2000 years interval. The intervals shorter than 100 years are all in 1 year step. We also evaluate the error rate of the model in each sliding window of 15, 20, 30, and 40 year lengths. The error rate is calculated by the number of invalid predictions that have significance $P > 0.05$ divided by the number of total predictions within a given window.

5. Comparison with Empirical Orthogonal Function analysis. We compare the predictability of $AGR_R$ by applying Ridge Regression and generalized linear model that uses the principal components of SLP fields estimated by Empirical Orthogonal Function (see, e.g., von Storch and Zwiers (1999)) decomposition as predictors. We first select three different DJF SLP spatial domains (global, $18°$ N–$18°$ S, and $18°$ N–$72°$ S). For each spatial domain, we select the first 10 components of SLP anomalies. When selecting the global SLP field, the first 10 components explain up to 75% of the SLP variance (see Appendix A Fig. A14). We then reconstruct SLP fields based on an increasing number of components, from 1 (based on component 1) to 10 components. These components are used as predictors of $AGR_R$ in a simple linear regression (also with leave–one–out cross–validation). The periods 1959–2017 and 1980–2017 are included (see Appendix A Fig. A13, A14).

## 3    Results and discussion

### 3.1    Global IAV patterns

In this section, we test the predictability of the global C-cycle IAV by using different predictors: global SLP fields by Ridge Regression, teleconnection indices by Ridge Regression, and the single SOI index by simple linear regression. In this context, each detrended annual $CO_2$ time–series (including $AGR_R$, $SL_{Resid}$, $SL_{DGVMs}$, $SL_{CAMS}$, and $SL_{CarboScope}$) is predicted by the above predictors over DJF, MAM, and DJF+MAM separately (Fig. 2). The predictability is evaluated by the Pearson correlation ($\rho$) between the original and predicted detrended annual $CO_2$ time–series. $\rho_{SLP}$, $\rho_{Tele}$, and $\rho_{SOI}$ represent the

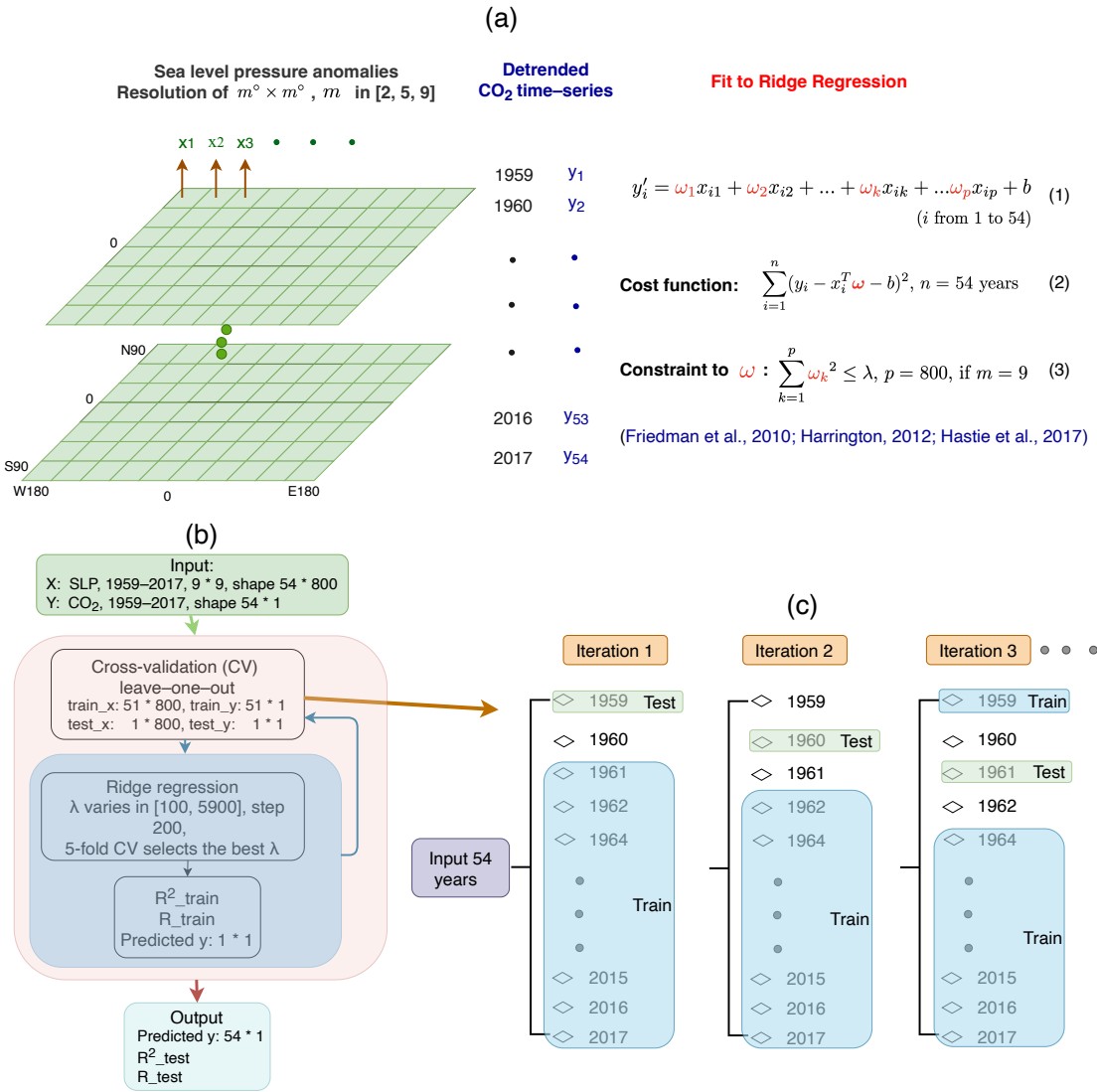

**Figure 1.** Schematic representation of the statistical approach and model design, with an example of the selected time interval of 1959–2017. (a) Fundamental principle of Ridge Regression. (b) Model training and validation under Ridge Regression leave–one–out cross–validation. (c) Train and test grouping through leave–one–out.

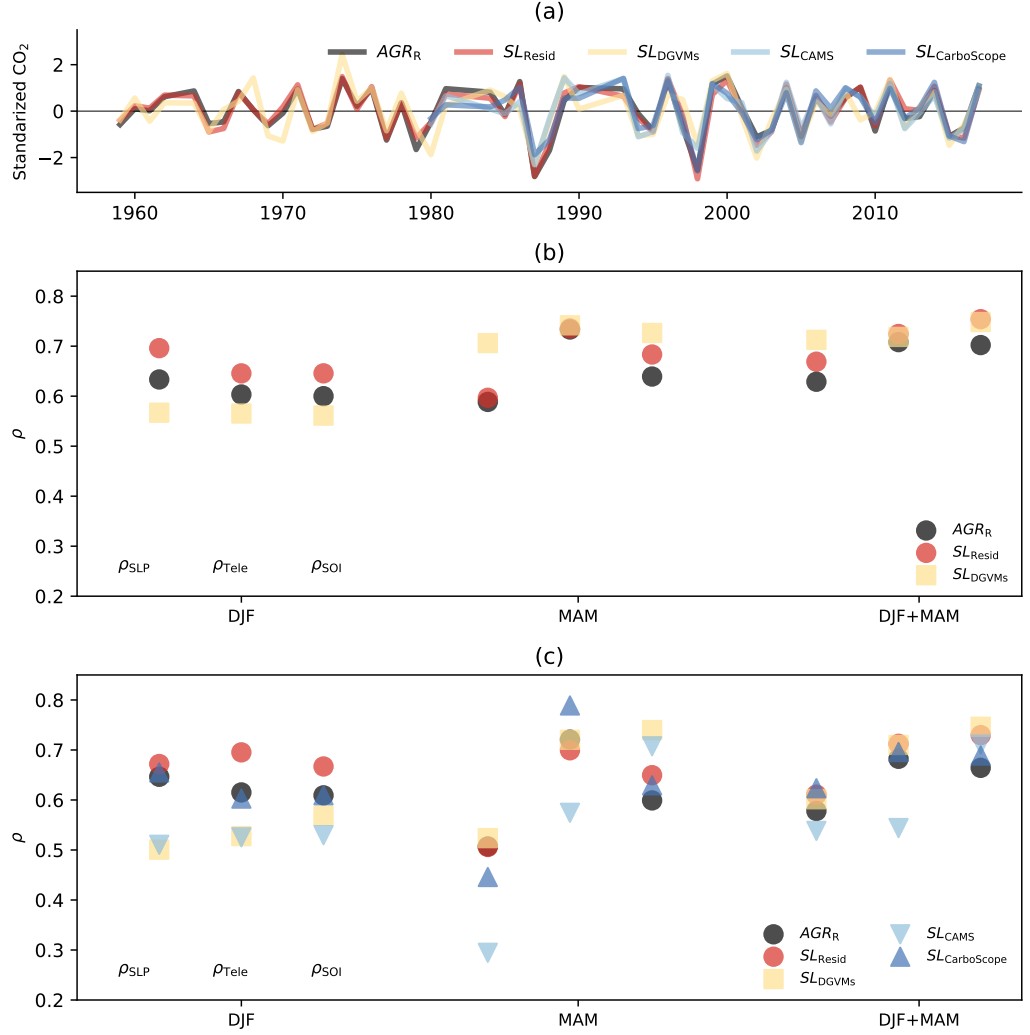

**Figure 2.** (a) Standardized annual observed/modeled $CO_2$ time–series over period 1959–2017 ($AGR_R$ in black, $SL_{Resid}$ in red and $SL_{DGVMs}$ in yellow), and in period 1980–2017 ($SL_{CAMS}$ in light blue and $SL_{CarboScope}$ in dark blue). The $CO_2$ time–series have all been detrended as described in Section 2. Note that the $AGR_R$, $SL_{Resid}$, and $SL_{DGVMs}$ in period 1980–2017 are detrended data based on their relevant period, and compared with detrended data based on 1959–2017, the difference is negligible. (b) Pearson correlation of predicted vs observed/modeled $CO_2$ time–series based on the Ridge Regression with SLP fields ($\rho_{SLP}$) or teleconnection indices ($\rho_{Tele}$) as predictors. Additionally, Pearson correlation of predicted vs observed/modeled $CO_2$ time–series by linear regression is based on the single predictor of SOI index ($\rho_{SOI}$). SLP fields, teleconnection indices, and SOI are aggregated for different seasons: DJF, MAM, and DJF+MAM. Panel (b) shows results for 1959–2017 and panel (c) for 1980–2017. Note that in panel (c), the $\rho_{SLP}$ of $SL_{CAMS}$ using MAM SLP as predictor has significance $P = 0.09$, all others have significance $P < 0.05$.

predictability by using different predictors of global SLP fields, teleconnection indices, and SOI respectively. Accordingly, the relevant Ridge Regression coefficients with above different predictors are represented as $\omega_{\text{SLP}}$, $\omega_{\text{Tele}}$, and $\omega_{\text{SOI}}$.

First, we find the detrended annual $CO_2$ time–series are generally consistent with each other, except the $SL_{\text{DGVMs}}$ shows slight deviation (Fig. 2 a). We find two anomalous years (1987 and 1998), which show deviations larger than 2 standard deviations in most $CO_2$ time–series, both signifying apparent $AGR$ increases and subsequent lower land sink (Fig. 2 a). These two years correspond to strong El Niño events, which are usually associated with below–average land $CO_2$ uptake (Keeling et al., 1995; van der Werf et al., 2004; Bonan, 2016).

Global SLP and teleconnection indices show comparable predictive skill of global C–cycle IAV in winter, while teleconnection indices have higher predictive skill in spring (Fig. 2 b). In both periods, the value of $\rho_{\text{SLP}}$ (except $SL_{\text{DGVMs}}$) is higher in DJF (0.51–0.70) than in MAM (0.29–0.60). On the other hand, the values of $\rho_{\text{Tele}}$ are higher in MAM (0.57–0.79) than in DJF (0.53–0.70). The relative low predictive skill of global SLP anomalies compared to teleconnection indices might result from: 1) limited sample size (less than 60 years) and a large number of predictors ($p = 800$) for Ridge Regression training with global SLP anomalies. But for teleconnection indices and SOI, the predictive skills are much less influenced by the limited sample size due to their limited predictors ($p \leq 15$ for teleconnection indices and $p = 1$ for SOI). As we increase the sample size to over 100 years, the predictive skill of SLP anomalies increases considerably, as is shown in temporal sensitivity study (Fig. 6), and 2) the predictive skill of SLP anomalies in explaining global C–cycle IAV can be reduced in domains with large local rather than global impacts of atmospheric variations to land carbon sinks (Jung et al., 2017). In such domains, the SLP anomalies might show strong relationship to local C–cycle variations but weaker link to global C–cycle variations. Selecting the SLP domains with higher contribution to the global C–cycle variability could improve the predictability, as is shown by the analyzes of sensitivity of the results to the SLP spatial domains (Fig. 4).

We find that the Ridge Regression using the full SLP fields as predictors yields general higher predictability than that of linear regression based on principal components of SLP (see Appendix A Fig. A13, A14). In winter and in the SLP domain of 18° N to 72° S, the linear regression based on the leading principal components of SLP shows lower predictability that the Ridge Regression, for different numbers of components retained (see Appendix A Fig. A13, $\rho < 0.7$ in 1959–2017 and $\rho < 0.75$ in 1980–2017). The lower predictive skill of the linear regression based on the leading components of SLP might be due to: **1)** principal component analysis captures the main variances of the SLP field, but not necessarily that of the main patterns that influence $CO_2$ IAV, **2)** mathematically, Ridge Regression and principal component analysis are deeply connected. Principal component analysis cuts off all components with small variance beyond a certain threshold, while Ridge Regression shrinks them, which allows for information in low variance components to be used for prediction (van Wieringen et al., 2021). Thus, Ridge Regression might reveal hidden components of SLP variability that are nevertheless important for $CO_2$ IAV.

Compared to the predictive skill of teleconnection indices, which includes a set of 14 teleconnection indices for period 1959–2017 and 15 for period 1980–2017 as predictors, the predictive skill of SOI is slightly lower or similar in both seasons, with 0.53–0.67 in DJF and 0.60–0.74 in MAM (Fig. 2 b). This is consistent with the dominant role of ENSO in driving global C–cycle IAV, with other modes showing less contributions. Such interpretation requires caution as the indices cannot fully represent the complex atmospheric dynamics.

The predictive skill of the combined winter and spring global SLP anomalies reveal the different seasonal responses of global C-cycle IAV to large–scale atmospheric circulation variability (Fig. 2 b). The predictive skill of SLP and teleconnection indices in DJF+MAM is within the values for DJF and MAM for most datasets, and slightly higher than the best performing season for the predictive skill of SOI.

The predictive skill of SLP to $SL_{Resid}$ is similar to $SL_{DGVMs}$ in MAM and slightly higher than $SL_{DGVMs}$ in DJF. The difference in the predictive skill of SLP to $SL_{Resid}$ and $SL_{DGVMs}$ in DJF may due to: 1) compared to $SL_{Resid}$, land sink IAV simulated by DGVMs is less sensitive to DJF climate forcing (Bastos et al., 2018), 2) $SL_{Resid}$ implicitly includes the variability from land use change as well as ocean sink variations (Dufour et al., 2013; DeVries et al., 2017; Friedlingstein et al., 2019).

We next compare the spatial patterns of the Ridge Regression coefficients of SLP and teleconnection indices in the period of 1959–2017, the results of the period 1980–2017 can be found in Appendix A Fig. A4 and A5. The spatial patterns of the $\omega_{SLP}$ are similar for the three $CO_2$ time–series: positive coefficients over eastern tropical Pacific Ocean and negative coefficients from Southeast Asia extending to Australia that together roughly consistent with ENSO, and negative from west Pacific (Fig. 3 a). In winter the positive coefficients over the eastern tropical Pacific are higher than in other regions, which are influenced by El Niño and La Niña respectively (Monahan, 2001; Hsieh, 2004; Rodgers et al., 2004; Schopf and Burgman, 2006; Sun and Yu, 2009; Yu and Kim, 2011): El Niño induces negative SLP anomalies over the East Pacific and positive SLP anomalies over the west Pacific (see King et al. (2020), Fig. 5). The results are consistent with the land sink being negatively driven by ENSO in winter: strong El Niño, decreased land sink, strong La Niña, increased land sink. In spring, the area over the central and western tropical Pacific also shows stronger coefficients, and likely corresponds to a mix of different modes, such as the ENSO, West Pacific teleconnection and the Interdecadal Pacific Oscillation, all showing strong coefficients in Fig. 3 b (SOI, WP and TPI indices). In Appendix A Fig. A11 we show the anomalies in temperature and precipitation associated to these patterns, as well as those in NBP from the two atmospheric inversions (see Appendix A Fig. A12). Generally, the temperature anomalies over the tropics show negative correlations to annual land sink (SLP driven $AGR_R$) in both winter (as high as –0.85) and spring (as high as –0.73), while weaker but positive correlations are found in Eurasia. Tropical precipitation anomalies show roughly positive correlations in winter (as high as 0.73) and in spring (as high as 0.67). This pattern indicates that $AGR_R$ is generally higher for cooler and wetter conditions over the tropics and Southern Hemisphere semi–arid regions in both seasons, which result in increased NBP (see Appendix A Fig. A12), as well as cooler but predominantly drier conditions over Eurasia, which result in a complex pattern of NBP anomalies (see Appendix A Fig. A12). These results are consistent with the strong ENSO fingerprint on the IAV of global $CO_2$ atmospheric growth rate and global land sink, e.g. as pointed out by Piao et al. (2020) and with the importance of southern semi-arid ecosystems (Ahlström et al., 2015), for IAV in the global land sink.

The higher Ridge Regression coefficients of teleconnection indices are consistent with the high sensitivity domains of the Ridge Regression coefficients of SLP corresponding to the patterns of ENSO and WP (Fig. 3 b). Our results show that C–cycle IAV reveals a high positive sensitivity to SOI (0.18 to 0.21) and negative sensitivity to TPI (-0.24 to -0.28) and WP (-0.09 to -0.15) in winter. High sensitivities are also found for DMI in winter (negative) and AMO in spring (negative) (Fig. 3 b). We find that the global C–cycle IAV is very sensitive to TPI as well as SOI, which is not so obvious for the spatial patterns of

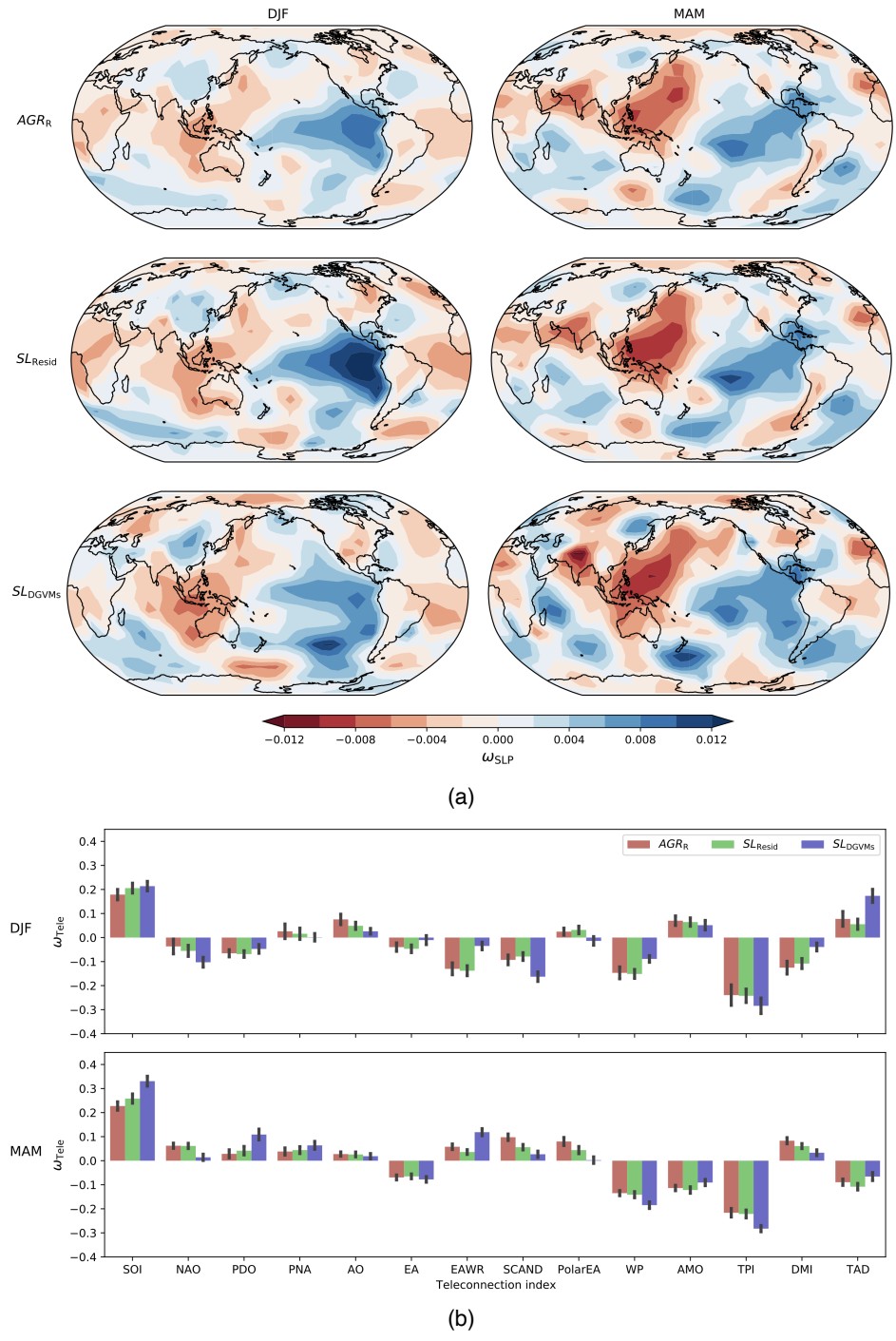

**Figure 3.** (a) Distribution of Ridge Regression coefficients of SLP with the time–series of $AGR_R$ (top row), $SL_{Resid}$ (center row) and $SL_{DGVMs}$ (bottom row) in DJF (left column) and MAM (right column) based on SLP fields in the period 1959–2017. (b) Distribution of Ridge Regression coefficients of teleconnection indices with $AGR_R$, $SL_{Resid}$ and $SL_{DGVMs}$. Both $\omega_{SLP}$ and $\omega_{Tele}$ are the mean of the $n = 54$ run Ridge Regression coefficients.

SLP Ridge Regression coefficients. TPI is a hemispheric–scale index and defined as the pressure anomaly differences between the locations Hobart (43° S, 147 °E) and Stanley (52° S, 58° W) (Pittock, 1980, 1984). We find the TPI to be strongly anti–correlated with SOI in winter and spring (-0.89 and -0.85, respectively). This might indicate an amplification of ENSO impacts on C–cycle IAV due to large–scale atmospheric circulation variability in the Southern Hemisphere.

However, the observed patterns of the Ridge Regression coefficients in teleconnection indices and SLP need to be interpreted with caution, since these patterns are not necessarily independent from each other. For example, the area from Southeast Asia extending to Australia corresponds to a region influenced by several large–scale atmospheric circulation modes: ENSO, the Indian Ocean Dipole (IOD), and the Southern Annular Mode (SAM) (Cleverly et al., 2016). Interactions between these modes have been shown to modulate the occurrence of drought and extreme precipitation in semi–arid areas of Australia, and thus induce large inter–annual variability in gross primary productivity in the region (Cleverly et al., 2016).

Compared to other $CO_2$ datasets, the predictability of $SL_{\mathrm{DGVMs}}$ is higher when using SLP fields in MAM as predictors, rather than DJF (Fig. 2 b). Moreover, $\omega_{\mathrm{SLP}}$ of $SL_{\mathrm{DGVMs}}$ exhibits distinct spatial patterns, especially in winter, where $\omega_{\mathrm{SLP}}$ for $SL_{\mathrm{DGVMs}}$ show higher values in the Southern Pacific rather than over the tropical Pacific region (Fig. 3 a). Compared to historical results, predictability of $SL_{\mathrm{DGVMs}}$ is lower by using winter SLP as predictors in Ridge Regression, rather than by using spring SLP, and the spatial patterns of the Ridge Regression coefficients for $SL_{\mathrm{DGVMs}}$ are slightly different. These differences might be an indication of shortcomings of DGVMs in simulating the sensitivity of land sink to climatic drivers.

The general match of spatial patterns of the Ridge Regression coefficients using SLP and the teleconnection indices as predictors of global C–cycle IAV indicates that SLP can capture the spatial distribution of the atmospheric patterns that influence IAV, with the advantage of being more flexible than teleconnection indices, since it does not require predefined definitions. However, the short sample size and the large number of predictors for Ridge Regression training hinder the performance of SLP anomalies, especially the lower predictability when using SLP anomalies than teleconnection indices in spring. Reducing the number of predictors (smaller domains of SLP anomalies) or increasing the sample size (longer time interval) for Ridge Regression training could improve the predictive skill of SLP anomalies. Therefore, in the next subsections, we conduct the spatial and temporal sensitivity study of the global C–cycle to SLP anomalies.

### 3.2 Sensitivity to the SLP domains

Here, we test the sensitivity of the global C-cycle IAV predictability by using different spatial domains of SLP as predictors in Ridge Regression. The SLP domains are selected over different latitudinal bands in DJF and MAM separately (Fig. 4). We find improved predictability in both seasons when selecting smaller spatial domains (particularly including the tropics to high latitudes of the Southern Hemisphere) rather than global SLP anomalies. MAM fields show lower predictability in general and are less sensitive to the spatial domain considered. In the following, we show the results for DJF, the results for MAM can be found in Appendix A Fig. A7. Here we only show the results of the period 1980–2017. The results of the period 1959–2017 show a similar trend (see Appendix A Fig. A6).

Consistent with previous studies (Zeng et al., 2005; Piao et al., 2020), the tropical domain corresponds to higher predictability for all datasets, but stronger predictability are found for regions extending from the tropics to the Southern Hemisphere

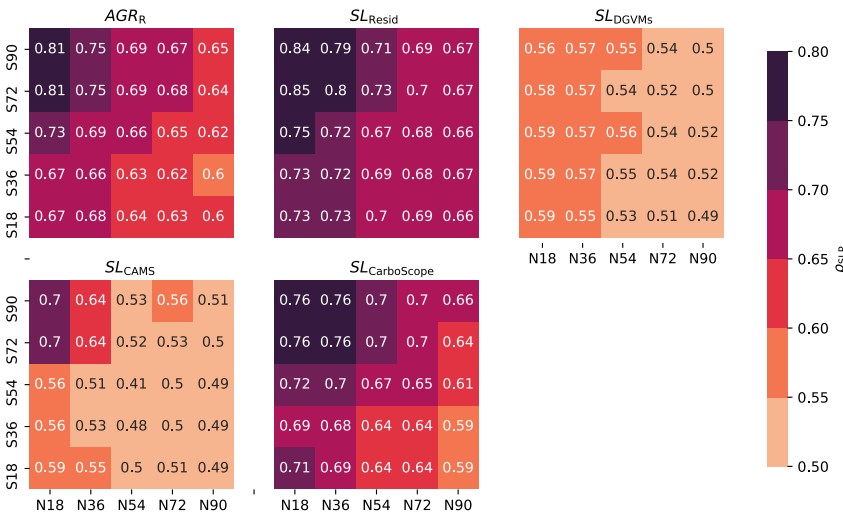

**Figure 4.** Heat map of predictability with $CO_2$ time–series over various SLP latitude domains in DJF. Each heat map contains 5×5 squares, and each square represents one domain of SLP. For example, the square 36° N–72° S is the domain of SLP extending from 36° N extending to 72° S. All latitudinal domains include the tropical area (18° N–18° S). The top right square thus represents global scale SLP. $\rho_{\mathrm{SLP}}$ of $AGR_{\mathrm{R}}$, $SL_{\mathrm{Resid}}$, $SL_{\mathrm{DGVMs}}$, $SL_{\mathrm{CAMS}}$, and $SL_{\mathrm{CarboScope}}$ in 1980–2017 are shown here.

(Fig. 4). Including Northern Hemisphere regions results in lower predictability. The domain 18° N–72° S shows the highest predictability, with $\rho_{\mathrm{SLP}}$ of 0.81 for $AGR_{\mathrm{R}}$ and 0.85 for $SL_{\mathrm{Resid}}$ in 1980–2017.

The results for net atmosphere-land fluxes estimated by atmospheric inversions are consistent with those of $AGR_{\mathrm{R}}$, with $\rho_{\mathrm{SLP}}$ of 0.70 for $SL_{\mathrm{CAMS}}$ and 0.76 for $SL_{\mathrm{CarboScope}}$ in the same domain of 18° N–72° S. The values of $\rho_{\mathrm{SLP}}$ of $SL_{\mathrm{DGVMs}}$ are systematically lower than the other datasets, independently of the domain.

    The weaker values of predictability when extending SLP domains from tropics to Northern Hemisphere (Fig. 4) might be due to the local rather than global impacts of large–scale atmospheric circulation variability in Northern Hemisphere to land sink IAV. Additional explanations include the fact that carbon fluxes are weaker in winter Northern Hemisphere, so that large–scale

atmospheric circulation variability exerts weaker influence in the global land sink, and that there are strong compensatory effects of gross primary productivity versus terrestrial ecosystem respiration in the Northern Hemisphere in response to water and temperature variations (Jung et al., 2017; Wang et al., 2022).

    The increasing predictability of global C-cycle IAV when using SLP fields extending from tropics to Southern Hemisphere (Fig. 4) is likely due to the strong contribution of semi–arid regions in the Southern Hemisphere extratropics to the global sink

through their drought/wet anomalies (Poulter et al., 2014; Ahlström et al., 2015). The drought/wet anomalies in these regions are controlled by large–scale atmospheric circulation variability in the Southern Hemisphere (ENSO and ENSO related modes)

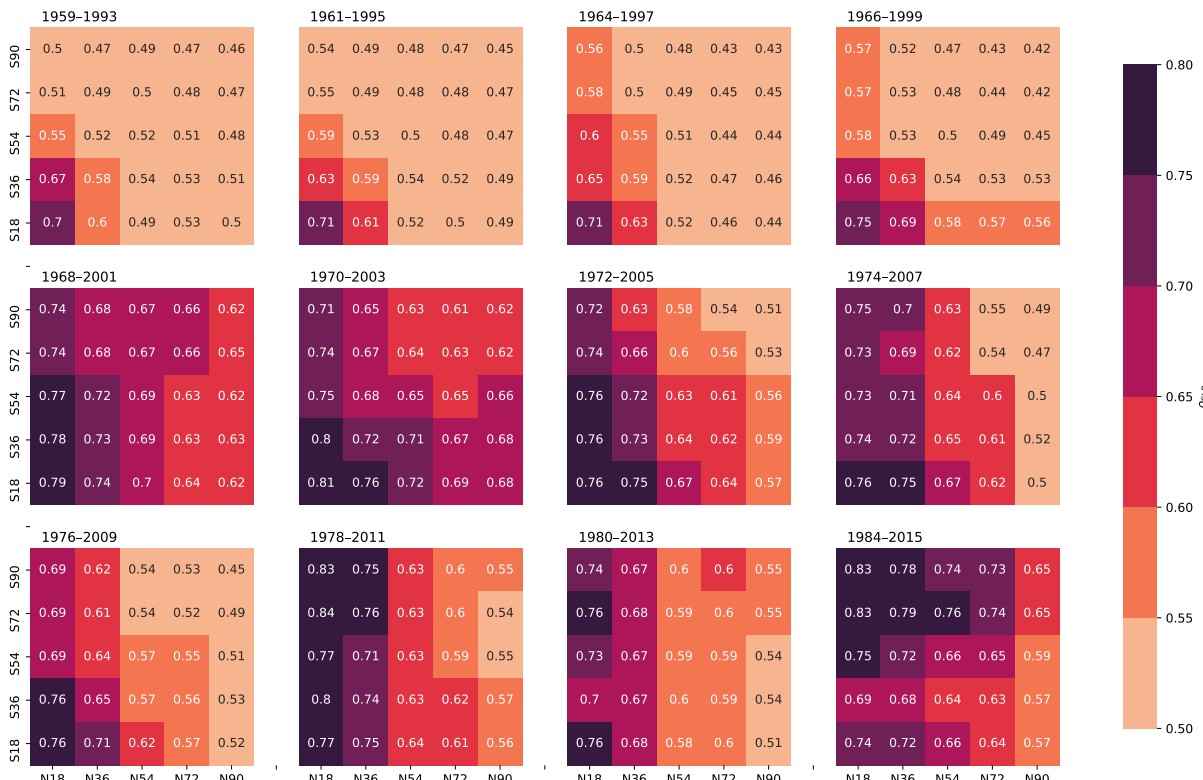

**Figure 5.** Predictability of $AGR_{\mathrm{R}}$ with DJF SLP over various latitude domains. A 30–yr sliding window in the period 1959–2017 with a one year step is created. The starting and end year of each interval is labeled on the top of each heat map. Here we only show the results of every second starting year, the full results are in Appendix A Fig. A9.

and to the interactions between ENSO and other large–scale atmospheric circulation modes in the Southern Hemisphere, such as the synergistic effects from ENSO, IOD and the SAM on Australia C–cycle variability (Cleverly et al., 2016).

### 3.3 Sensitivity to the temporal domains

Because of multi–decadal variability in the climate system, it is possible that the relationships found for short intervals are not stable. In order to investigate whether these results depend on the temporal domain considered, we additionally analyze the influence of different temporal domains (30–yr interval sliding window) on the predictability of global C-cycle IAV by selecting different temporal domains, and performing the Ridge Regression separately for the global and the tropical domains (Fig. 5).

Results show stronger predictability of $AGR_{\mathrm{R}}$ confined to the tropics of SLP domains in earlier periods and an intensification of these predictability for the SLP domains extending to the Southern Hemisphere over the study period. In some periods, the tropics and Southern extratropics domain shows the highest values of $\rho_{\mathrm{SLP}}$, for example in 1978–2011 and 1984–2015 (Fig. 5).

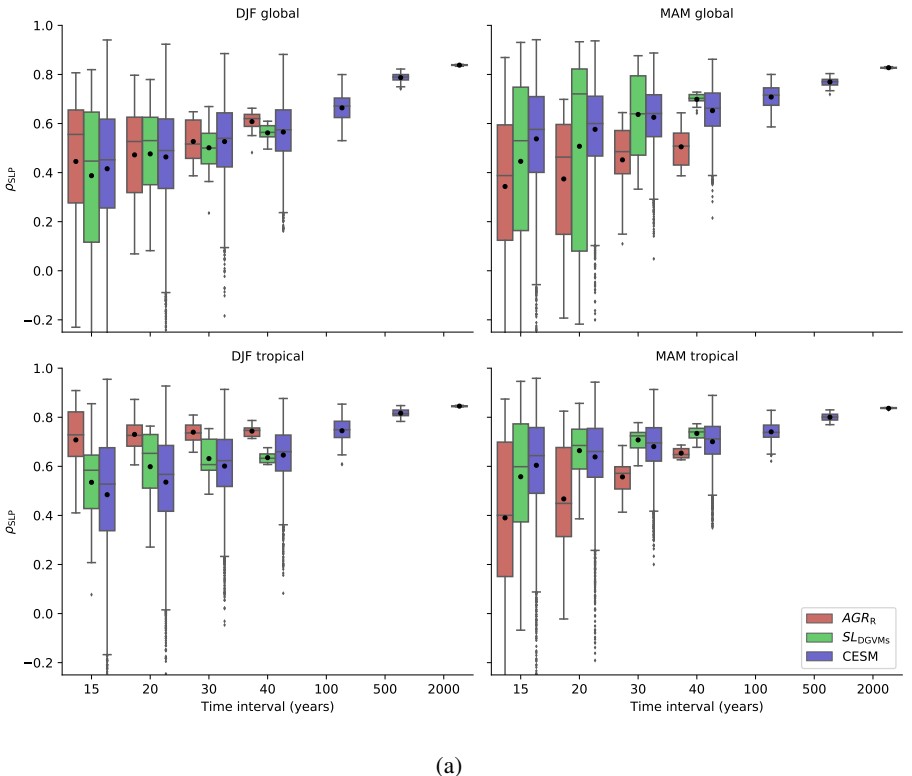

(a)

**Figure 6.** Predictability of $AGR_R$, $SL_{DGVMs}$, and CESM NBP under various time intervals. The predictability of $AGR_R$ and $SL_{DGVMs}$ are both within the period 1959–2017, with a 1 year step sliding window of 15, 20, 30, and 40 year. The predictability of CESM in the period 1000–5000 and covers extra intervals of 100, 500, 2000 years. The distribution of the predictability under each sliding window with SLP in DJF global, MAM global, DJF tropical and MAM tropical. Tropical domains are 18° N–18° S for SLP as predictors in predicting $AGR_R$ and $SL_{DGVMs}$, and 20° N–20° S for SLP as predictors in predicting CESM. Note that the mean values are in black dots.

There is, however, high temporal variability in the predictability and of the most relevant spatial domain, with other periods showing higher global coherence (e.g. 1968–2001). It is unclear whether these temporal variations occur randomly due to internal variability in the climate system, or are influenced by external forcing. Potential explanations for this pattern include trends found in SLP variability over the Pacific and Southern Atlantic (Schneider et al., 2012; IPCC, 2013; Roxy et al., 2019), or enhanced sensitivity of C–cycle variability to climatic drivers, particularly in semi–arid areas, under progressive climate change (Wang et al., 2014; Poulter et al., 2014).

Understanding and attributing these changes to given processes is beyond the scope of this study, but these results highlight the importance of the temporal domain when analyzing IAV in the global C–cycle. Since the observed $CO_2$ time–series are short and cover only limited temporal domains, results are likely to be affected by multi–decadal internal climate variability, in addition to external forcing. Moreover, the data–driven Ridge Regression method to quantify circulation–induced global

C–cycle variability uses a large number of predictors, while only relatively short time series are available for training, which may negatively affect the model's performance. Therefore, we further test the sensitivity of the predictability to the length of the time–series (Fig. 6). We test the predictability of global C–cycle IAV for different lengths of the temporal domain: 15, 20, 30 and 40 years for the datasets in Global Carbon Budget 2018 and CESM simulations and 100, 500 and 2000 years for CESM only.

The boxplots in Fig. 6 show the distribution of predictability calculated for multiple time intervals, each time interval using a sliding window over the whole period of the respective time–series of $AGR_R$, $SL_{DGVMs}$ and CESM. The spread of predictability provides an indication of internal variability in the predictability of global C–cycle IAV due to the choice of temporal domain and the uncertainty in the Ridge Regression fit for a large number of predictors and comparatively small number of training samples.

We find that the longer the time interval the higher the mean predictability and the smaller the variation, i.e. the less dependent are the results on the temporal domain considered (Fig. 6). However, the mean value tends to be lower than the median for intervals shorter than 30 years, and similar to the median for longer intervals. The lower mean is influenced by some domains with very low or even negative predictability from invalid predictions in shorter time intervals.

The mean predictability of $AGR_R$ in winter for the global domain increases from 0.45 to 0.61 from 15–yr to 40–yr respectively, while the spread (maximum $\rho_{SLP}$ - minimum $\rho_{SLP}$) decreases from 1.04 to 0.18. Predictability for $SL_{DGVMs}$ are consistent with those of $AGR_R$, with systematically lower mean predictability for winter and higher for spring, but similar spread in both. The mean value of predictability for CESM in SLP DJF over the global domain increases from 0.42 (15–yr) to 0.57 (40–yr) and to 0.84 (2000–yr), and the spread decreases from 1.72 to 0.72 and to 0.008, respectively.

At global scale, the predictive skill of SLP anomalies with $AGR_R$ and with models from $SL_{DGVMs}$ and CESM are different in winter and spring (Fig. 6). Predictability of $AGR_R$ is higher with winter SLP ($\rho_{SLP}$ is 0.61 in 40–yr DJF), but predictability of $SL_{DGVMs}$ and CESM are higher with spring SLP ($\rho_{SLP}$ in 40–yr MAM is 0.70 and 0.65, respectively).

When limiting the SLP domain to the tropics, results follow the same patterns as those at the global scale, but with better predictive skill: $AGR_R$ shows the highest mean predictability of 0.74 for intervals of 40 years in winter (Fig. 6). $SL_{DGVMs}$ shows the highest mean predictability, with $\rho_{SLP}$ of 0.73 for 40–yr spring (Fig. 6), a result that is very similar to those of CESM for the same temporal length ($\rho_{SLP}$ = 0.70). We find that with different time intervals, tropical SLP in winter leads to higher predictability of $AGR_R$ than global SLP fields. While the spring tropical SLP only shows slightly higher predictive skill than global SLP for $SL_{DGVMs}$ and CESM. $AGR_R$ is highly influenced by winter tropical SLP, while $SL_{DGVMs}$ and CESM are more sensitive to spring tropical SLP (Fig. 6). This is consistent with the results of Fig. 4 even when different time intervals are considered. This might be due to that Earth System Models, as CESM used here, have been found to not reliably simulate the seasonal timing of ENSO occurrence (Sheffield et al., 2013). The predictive skill of SLP in seasonal and domain differences between observation-based data and models ($SL_{DGVMs}$ and CESM) could be used as an indicator to reveal their different driving mechanisms.

We evaluate the error rate of valid predictions within each time interval sliding window in Fig. 6 (i.e., the fraction of predictions in one time interval sliding window with $\rho_{SLP}$ of significance $P > 0.05$) for $AGR_R$, $SL_{DGVMs}$, and CESM (see

Appendix A Fig. A10). We find that at least 30–year long intervals are needed for robust prediction of global C-cycle IAV. For periods shorter than 30 years, the rate of invalid predictions (in the sense given above) can be higher than 40 % for most datasets and SLP domains and seasons. It is worth noting that even when predicting $AGR_R$ by using winter SLP in the tropical domain, the error rate can still be as low as 13% in 15–yr interval. From the 15–yr to 30–yr interval, the error rate of $AGR_R$ reduces from 0.4 to 0 in winter global and 0.13 to 0 in winter tropical. The error rate decreases to less than 0.16 in 30–yr interval, except $AGR_R$ decreases to 0.24 in spring global, which also matches the relative low predictability of spring SLP anomalies in the period 1980–2017 with different spatial domains (see Appendix A Fig. A7 b). All error rates are reduced to almost zero in a 40–yr interval.

## 4   Conclusions

The major objective of this study is to explore the relationship between SLP anomalies (as a proxy of large–scale atmospheric circulation variability) and the global C–cycle IAV. Specifically, our goals are 1) to investigate the skill of SLP to predict global C–cycle IAV using Ridge Regression and to compare with traditional teleconnection indices, and 2) to establish statistical links at different spatio–temporal scales between large–scale atmospheric circulation variability and global C-cycle IAV.

First, we find that boreal winter and spring SLP anomalies allow predicting IAV of atmospheric $CO_2$ growth rate and IAV of the global land sink, with correlations between predicted and reference values between 0.70–0.84 when using winter SLP fields. This is comparable or higher than the predictive skill of a similar model using 15 teleconnection indices as predictors. The spatial patterns of the Ridge Regression coefficients reveal a strong influence of large–scale atmospheric circulation variability on global C–cycle IAV, particularly in the El Niño / Southern Oscillation and West Pacific domains. Second, the comprehensive spatio–temporal sensitivity analysis indicates an increasing sensitivity of global C–cycle IAV to large–scale atmospheric circulation variability during boreal winter in the Southern Hemisphere extratropics in the recent decades. This increased sensitivity may be influenced by internal climate variability or by enhanced sensitivity of global C–cycle variability to externally forced changes, but requires further research. Finally, we find that time–series of at least 30 years are needed for robust predictability of global C–cycle IAV. For shorter time–series, predictability is highly dependent on the particular period considered, and thus largely due to statistical artifacts of random internal climate variability in the fitting process.

Overall, Ridge Regression using seasonal SLP fields as predictors of global C-cycle variability provides a novel and efficient data-driven approach for detecting the relationship of large–scale atmospheric circulation variability to global C–cycle variability. Compared to teleconnection indices, this approach requires no pre–defined spatial configurations and is more flexible to the particular domain considered and shows equal or higher predictability of global C-cycle IAV. This method allows quantifying the contribution of atmospheric dynamical processes in driving variability in the C-cycle at global and regional scales, and may further be useful for attributing observed changes to internal climate variability versus anthropogenic climate change.

**Appendix A**

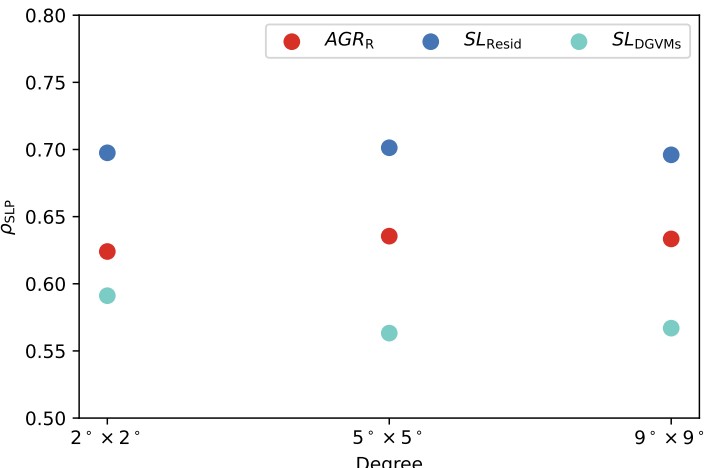

**Figure A1.** $\rho_{\mathrm{SLP}}$ of $AGR_{\mathrm{R}}$, $SL_{\mathrm{Resid}}$ and $SL_{\mathrm{DGVMs}}$ under different DJF SLP resolution ($2°{\times}2°$, $5°{\times}5°$, $9°{\times}9°$) by Ridge Regression leave–one–out cross–validation in period of 1959–2017.

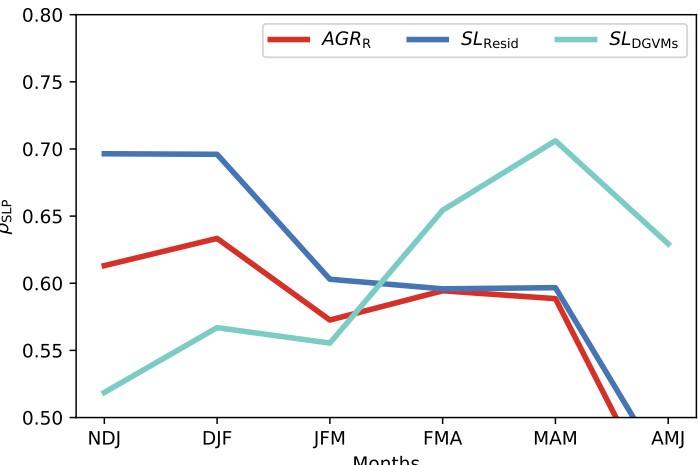

**Figure A2.** $\rho_{\mathrm{SLP}}$ of $AGR_{\mathrm{R}}$, $SL_{\mathrm{Resid}}$ and $SL_{\mathrm{DGVMs}}$ under different seasonal SLP (with different month combination) by Ridge Regression leave–one–out cross–validation in period of 1959–2017. Each combination represents: NDJ (November, December, and January), DJF (December, January, and February), JFM (January, February, and March), MAM (March, April, and May), AMJ (April, May, and June).

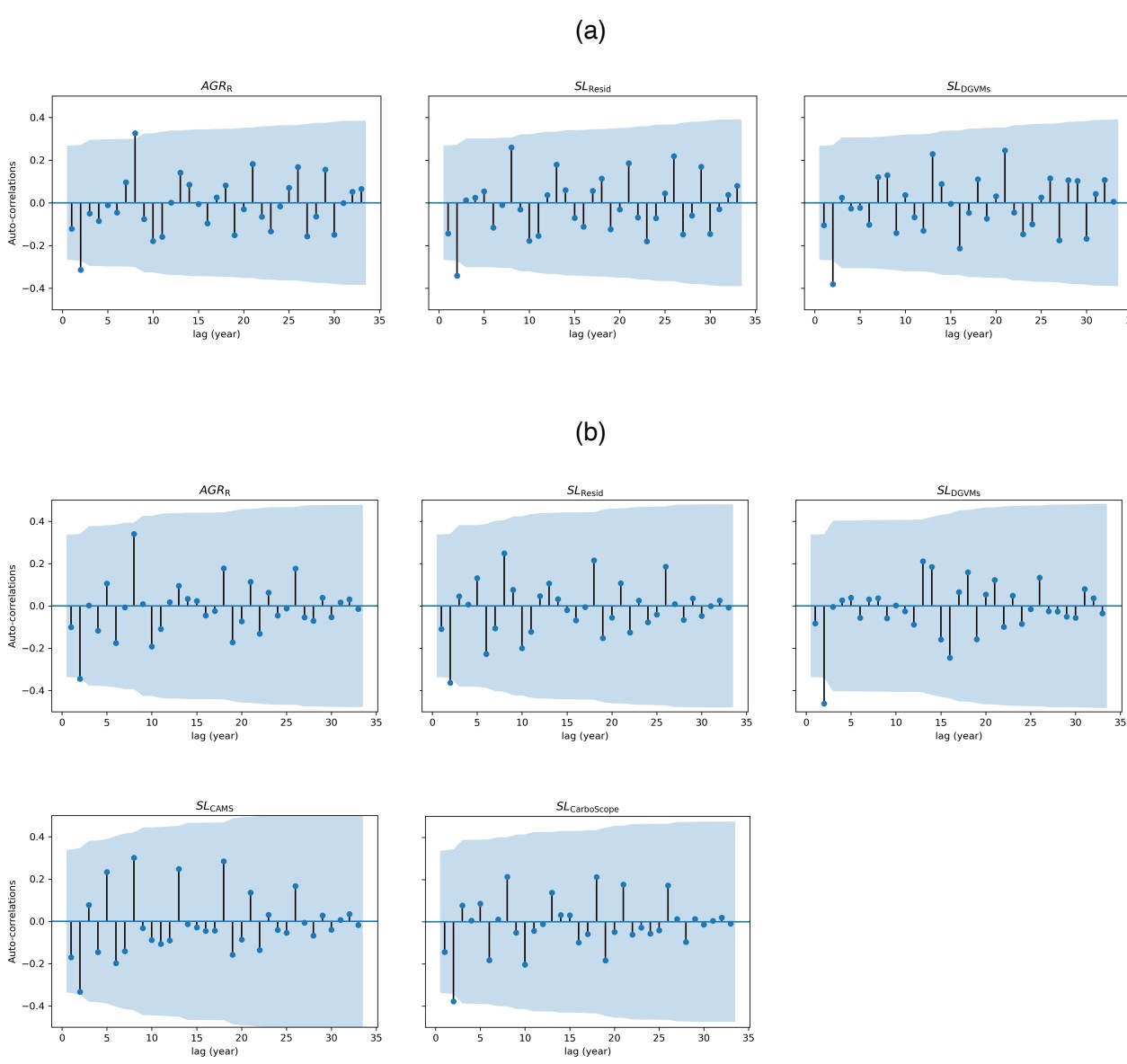

**Figure A3.** Time series auto–correlations of pre–treated $CO_2$ time–series in period (a) 1959–2017 for $AGR_{\mathrm{R}}$, $SL_{\mathrm{Resid}}$ and $SL_{\mathrm{DGVMs}}$. (b) 1980–2017 included two more inversions $SL_{\mathrm{CAMS}}$ and $SL_{\mathrm{CarboScope}}$. The shaded areas are the 95% confidence interval of the calculated auto–correlation under different lags.

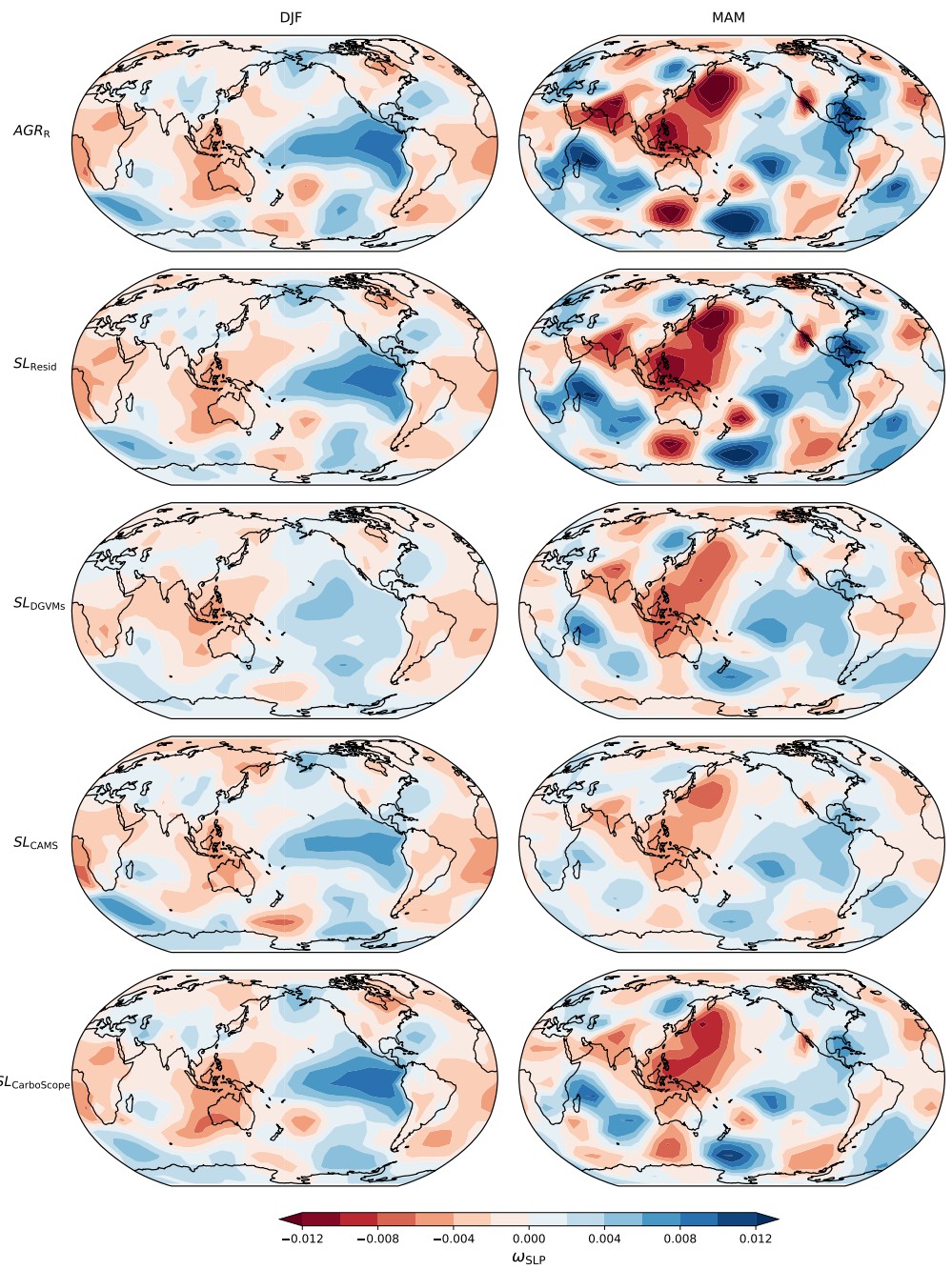

**Figure A4.** Distribution of $\omega_{\mathrm{SLP}}$ with the time–series of $AGR_{\mathrm{R}}$ (top row), $SL_{\mathrm{Resid}}$ (second row), $SL_{\mathrm{DGVMs}}$ (third row), $SL_{\mathrm{CAMS}}$ (fourth row) and $SL_{\mathrm{CarboScope}}$ (last row) in DJF (left column) and MAM (right column) based on SLP fields in the period 1980–2017. $\omega_{\mathrm{SLP}}$ are the mean of the $n = 34$ run Ridge Regression coefficients. Note that $\rho_{\mathrm{SLP}}$ of $SL_{\mathrm{CAMS}}$ MAM has $P > 0.5$.

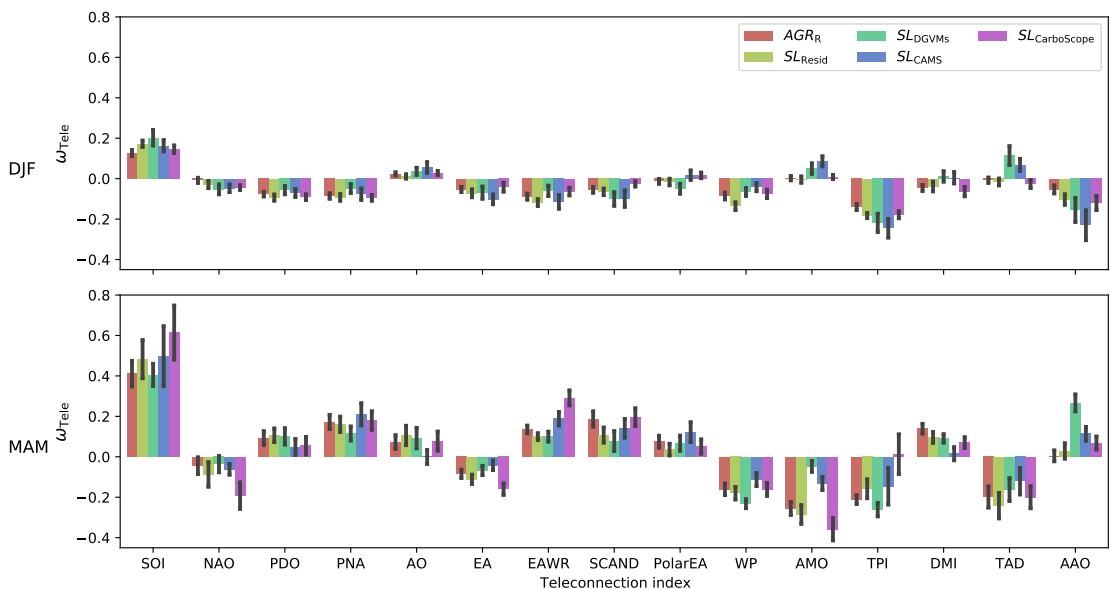

**Figure A5.** Distribution of $\omega_{\text{Tele}}$ with the time–series of $AGR_{\text{R}}$, $SL_{\text{Resid}}$, $SL_{\text{DGVMs}}$, $SL_{\text{CAMS}}$ and $SL_{\text{CarboScope}}$ based on teleconnection indices in period of 1980–2017.

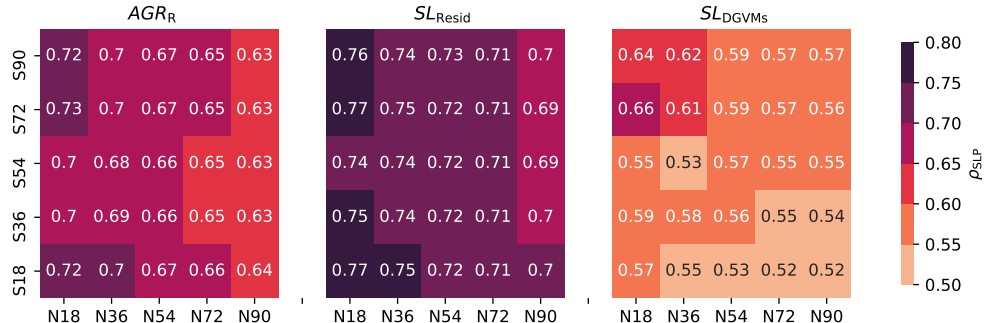

**Figure A6.** Heat map of $\rho_{\text{SLP}}$ with $CO_2$ time–series over various SLP latitude domains in DJF. Each heat map contains 5×5 squares, and each square represents one domain of SLP. For example, the square $36°$ N–$72°$ S is the domain of SLP extending from $36°$ N to $72°$ S. All latitudinal domains include the tropical area ($18°$ N–$18°$ S). The top right square thus represents global scale SLP. $\rho_{\text{SLP}}$ of $AGR_{\text{R}}$, $SL_{\text{Resid}}$, $SL_{\text{DGVMs}}$ in 1959–2017.

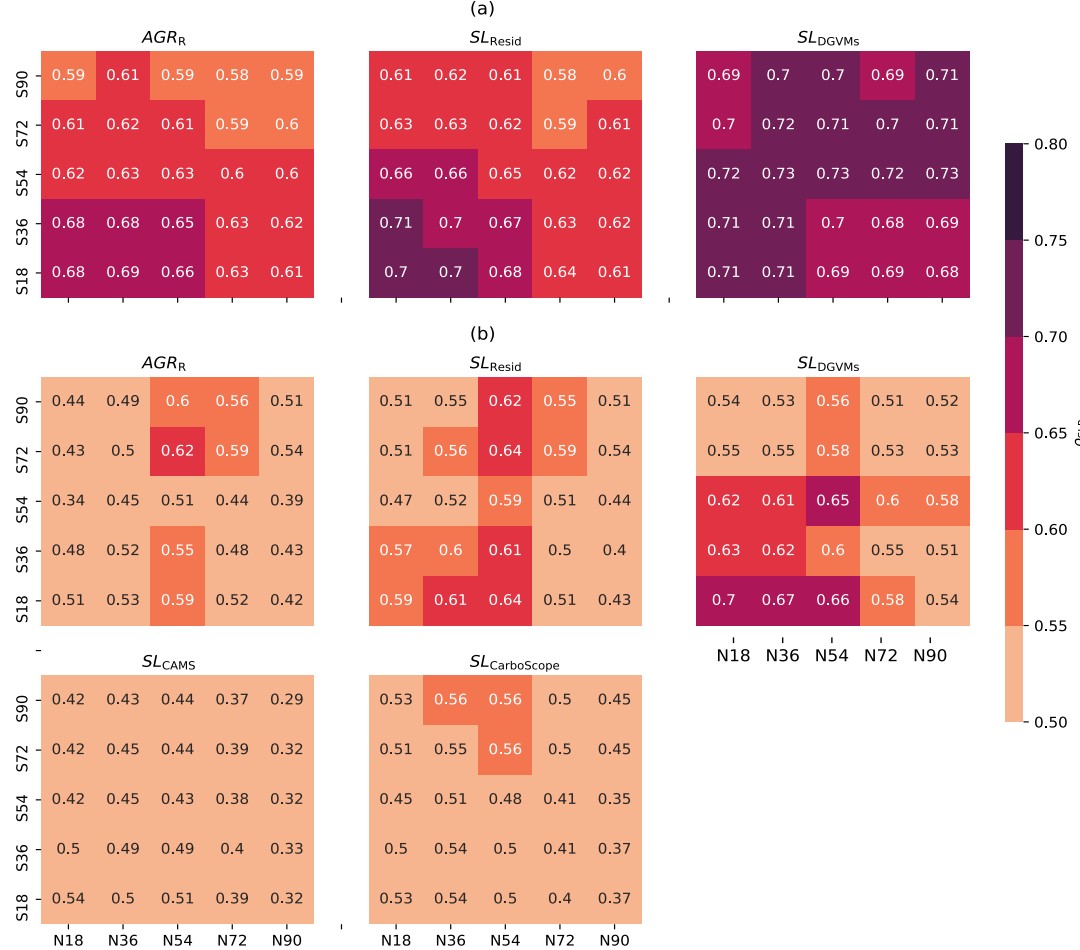

**Figure A7.** Heat map of $\rho_{\mathrm{SLP}}$ with $CO_2$ time–series over various SLP latitude domains in MAM. Each heat map contains 5×5 squares, and each square represents one specific domain of SLP. For example, the square 36° N–72° S is the domain of SLP extending from 36° N extending to 72° S. All latitudinal domains include the tropical area (18° N–18° S). The top right square thus represents global scale SLP. Time series is from (a) 1959–2017 for $AGR_{\mathrm{R}}$, $SL_{\mathrm{Resid}}$ and $SL_{\mathrm{DGVMs}}$. (b) 1980–2017 included two more inversions $SL_{\mathrm{CAMS}}$ and $SL_{\mathrm{CarboScope}}$.

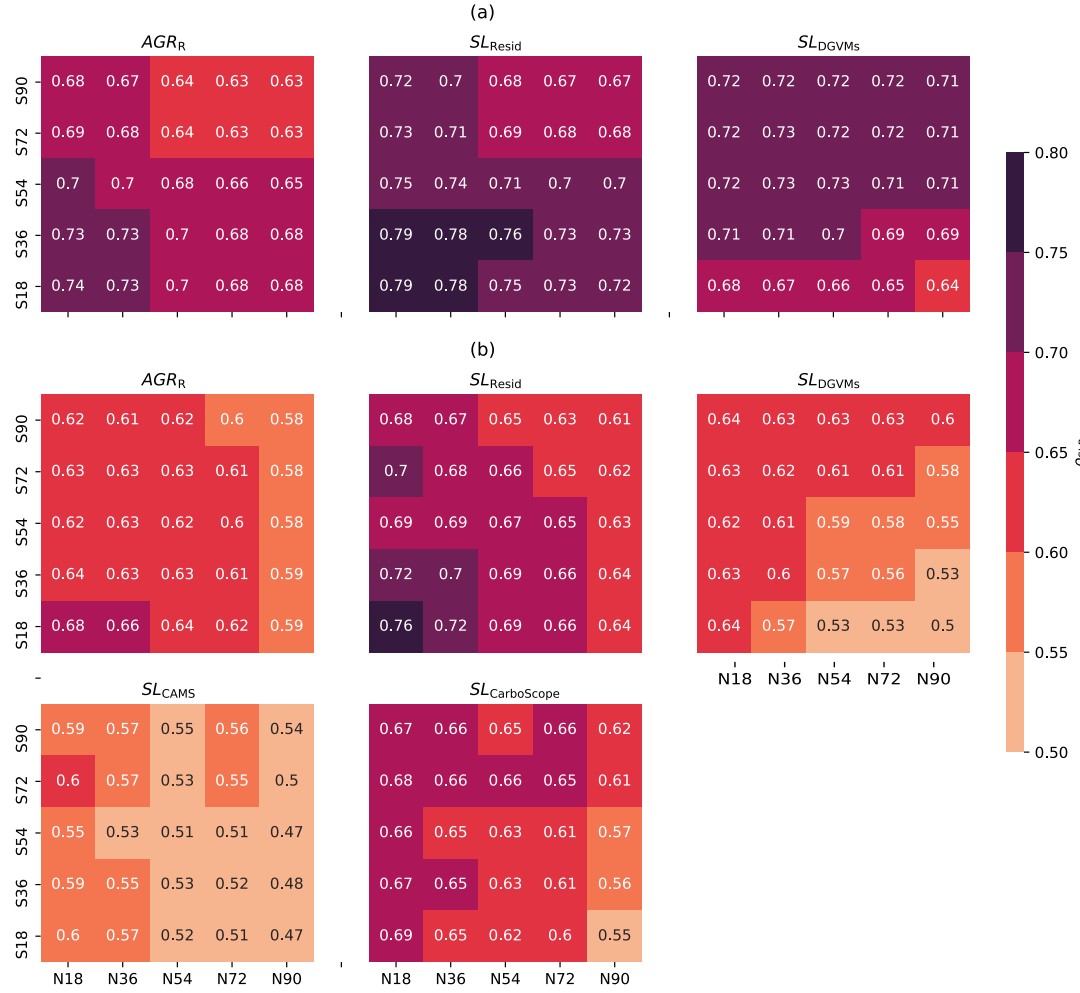

**Figure A8.** Heat map of $\rho_{\mathrm{SLP}}$ with $CO_2$ time–series over various SLP latitude domains in DJF+MAM. Each heat map contains 5×5 squares, and each square represents one specific domain of SLP. For example, the square 36° N–72° S is the domain of SLP extending from 36° N extending to 72° S. All latitudinal domains include the tropical area (18° N–18° S). The top right square thus represents global scale SLP. Time series is from (a) 1959–2017 for $AGR_{\mathrm{R}}$, $SL_{\mathrm{Resid}}$ and $SL_{\mathrm{DGVMs}}$. (b) 1980–2017 included two more inversions $SL_{\mathrm{CAMS}}$ and $SL_{\mathrm{CarboScope}}$.

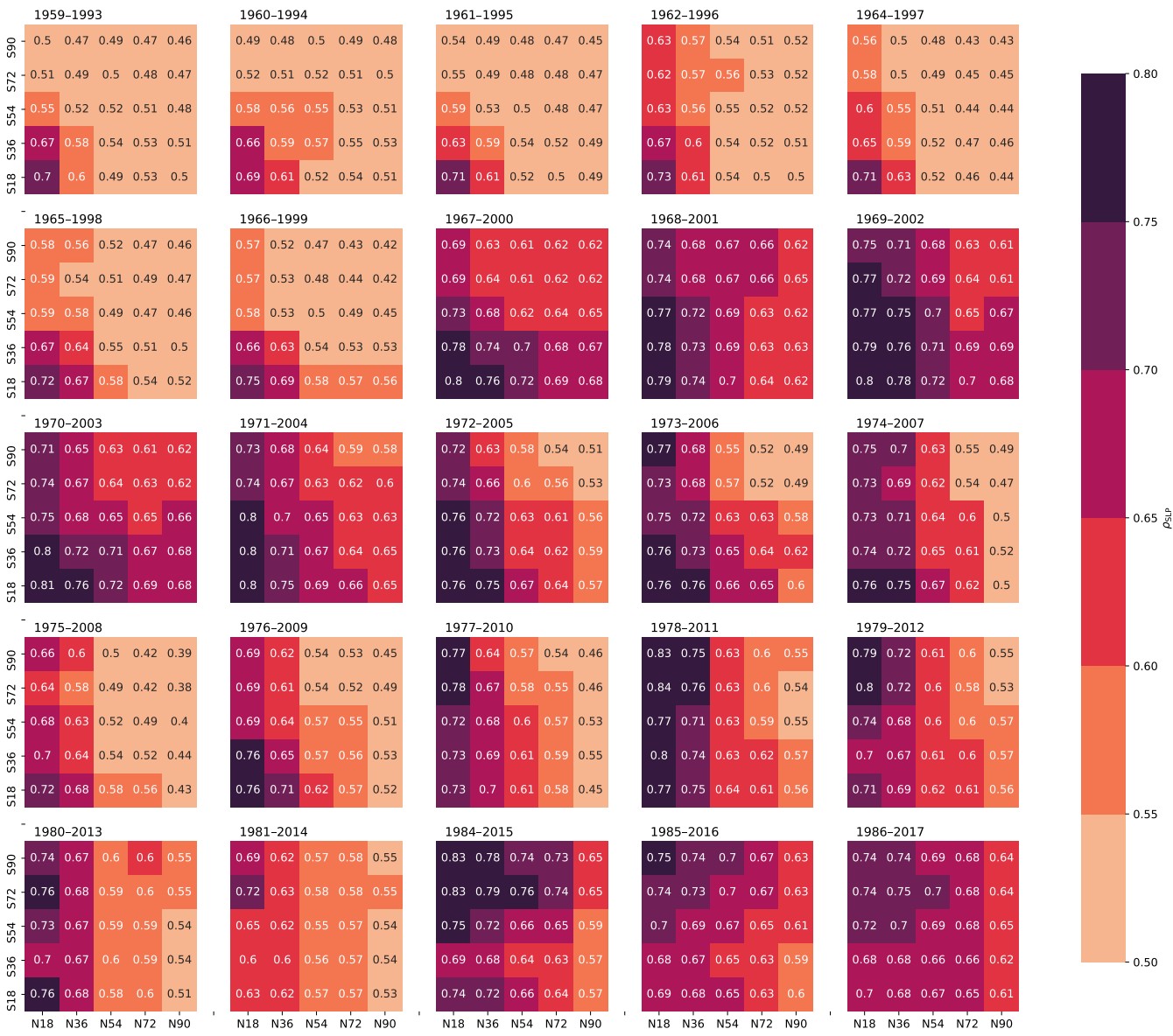

**Figure A9.** Heat map of $\rho_{\mathrm{SLP}}$ of $AGR_{\mathrm{R}}$ with DJF SLP over various latitude domains. A 30–yr sliding window in the period of 1959–2017 with a one year step is created. The starting and end year of each interval is labeled on the top of each heat map.

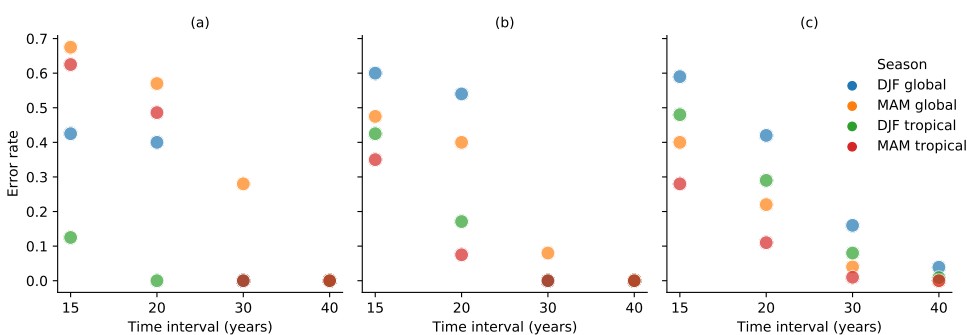

**Figure A10.** Error rate of the sliding window in 15, 20, 30 and 40 year intervals. For each sliding window, the error rate is calculated by the number of invalid predictions (with significance $P > 0.05$ in $\rho_{\mathrm{SLP}}$ of $CO_2$ time–series) divided by the number of total predictions. With SLP in DJF global, MAM global, DJF tropical and MAM tropical as predictors, the error rate of $\rho_{\mathrm{SLP}}$ of (a)$AGR_{\mathrm{R}}$, (b) $SL_{\mathrm{DGVMs}}$, and (c) CESM are plotted.

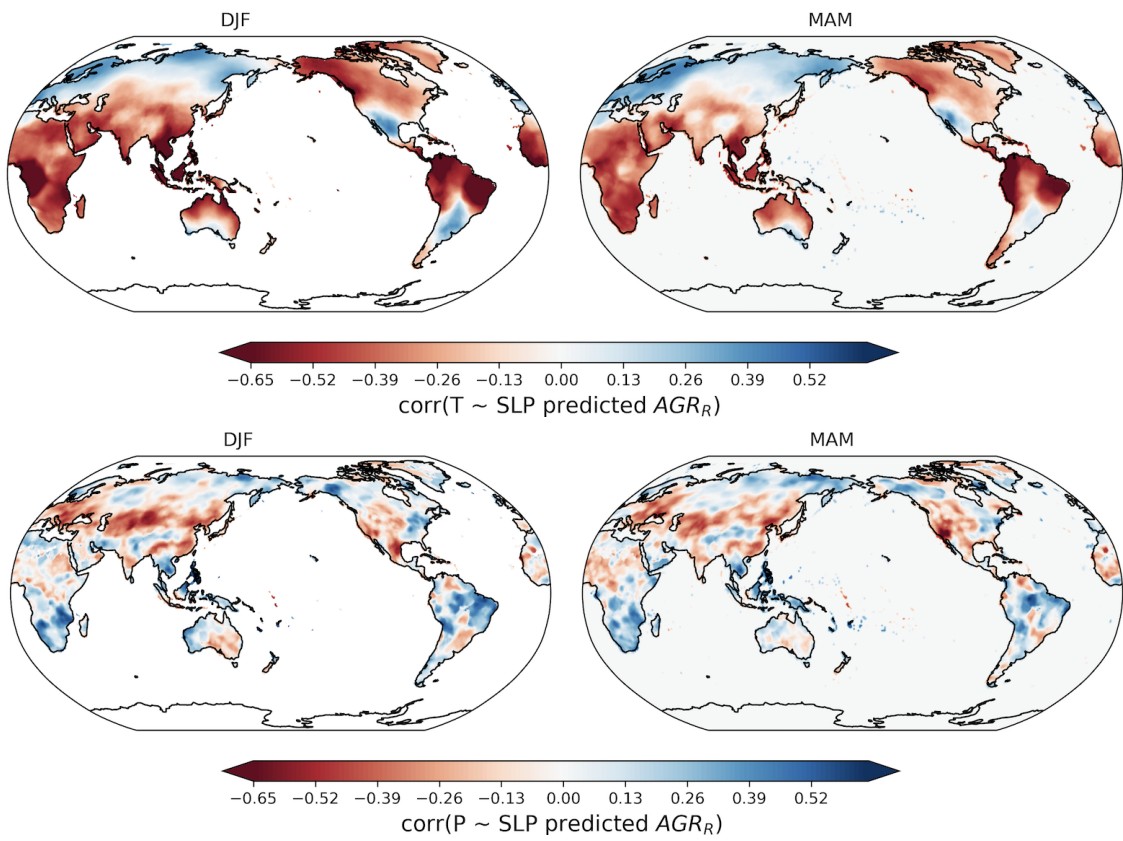

**Figure A11.** The spatial distribution of Pearson correlations between global SLP predicted $AGR_R$ (one time–series) to global pixel–based land temperature/precipitation anomalies (both from CRU TS4.05 (Harris et al., 2020) monthly dataset, aggregated to annual mean temperature/annual sum precipitation, and detrended by LOWESS), in the period 1980–2017. The top panel shows the spatial distribution of Pearson correlations between pixel–based land temperature anomalies to global SLP predicted $AGR_R$ for DJF (left) and MAM (right), and the bottom panel shows Pearson correlations of land precipitation anomalies.

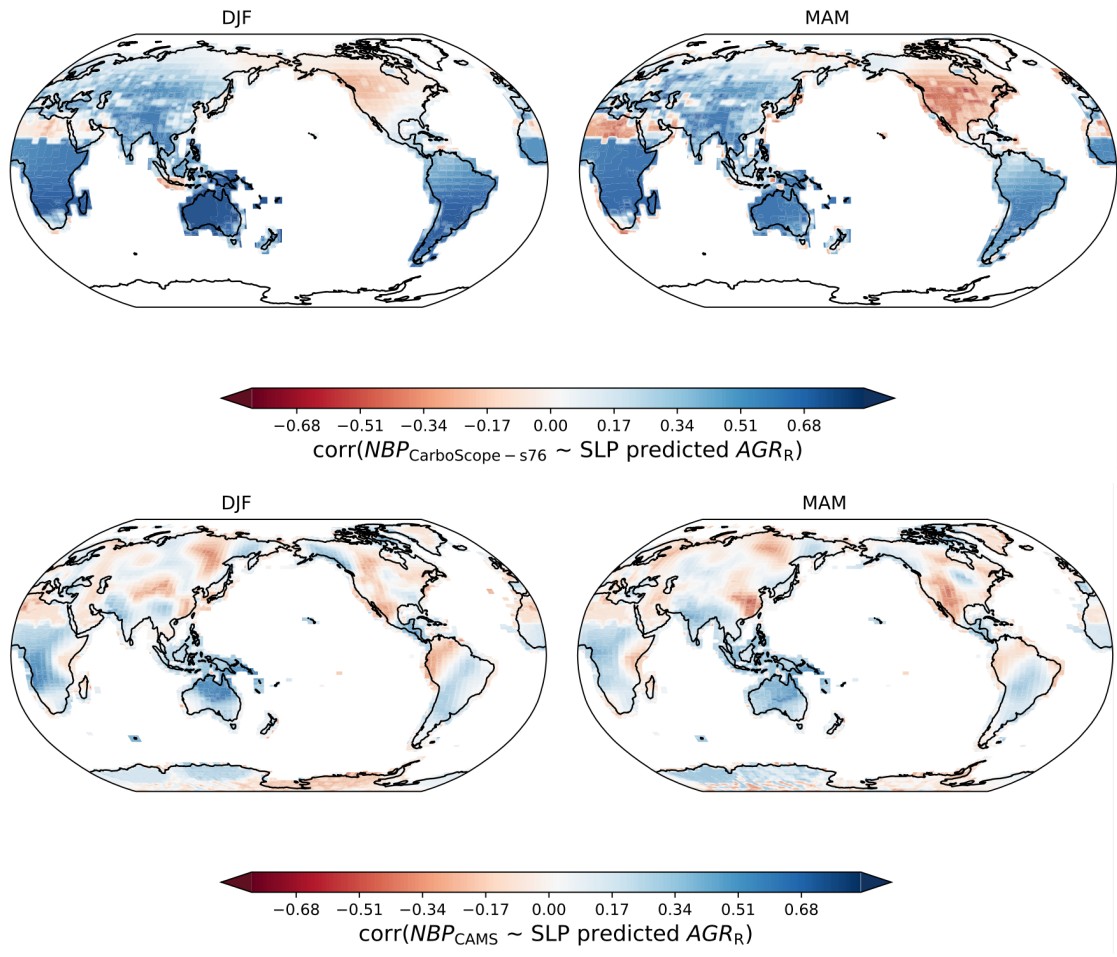

**Figure A12.** The spatial distribution of Pearson correlations between global DJF/MAM SLP predicted $AGR_R$ (one time–series) with pixel–based annual sum NBP variation (LOWESS detrended) from atmospheric inversion CarboScope s76 (Bastos et al., 2020; Chevallier et al., 2005; Rödenbeck et al., 2003) (upper panel) and CAMS (Bastos et al., 2020; Chevallier et al., 2005; Rödenbeck et al., 2003) (lower panel), in the period 1980–2017.

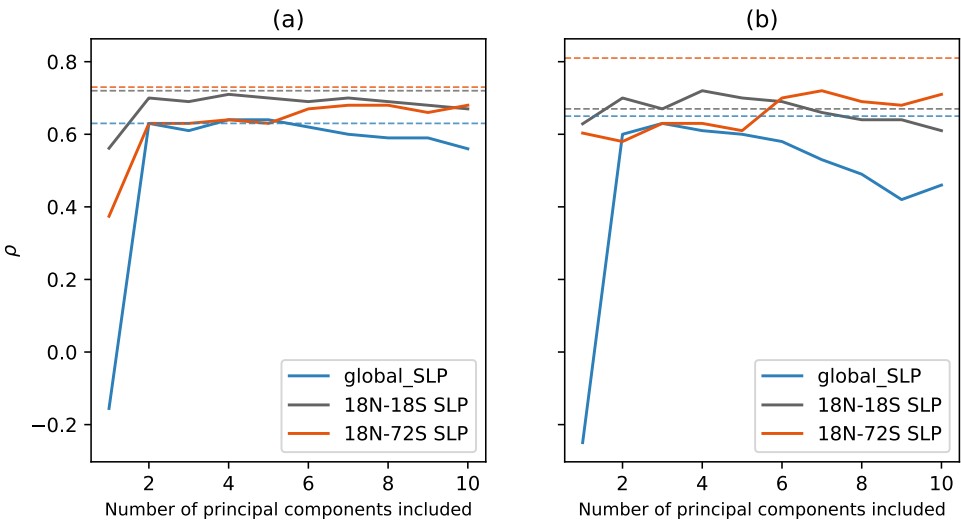

**Figure A13.** Comparison of linear regression based on SLP principal components to Ridge Regression based on SLP fields in the period of (a) 1959–2017 and (b) 1980–2017. The label of y axis shows the Pearson correlation between the predicted to original $CO_2$ time–series. The solid lines represent the results by using the linear regression based on different number of SLP principal components as predictors. The dashed lines represent the results by using Ridge Regression. Different colored lines represent different SLP spatial domains selected.

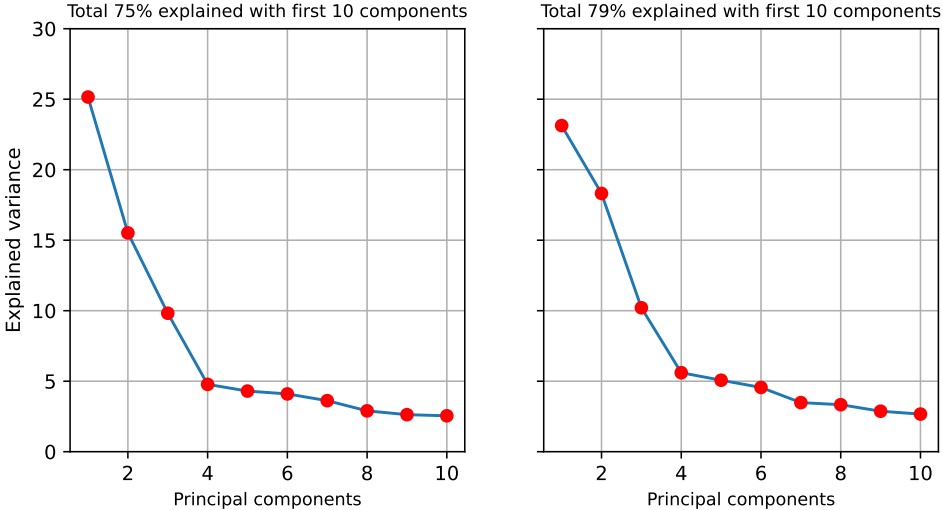

**Figure A14.** The explained variance of SLP principal components that extracted from global DJF SLP fields, in the period of 1959–2017 (left panel) and 1980–2017 (right panel).

## Appendix B: Data availability

The Global Carbon Budget 2018 dataset (Le Quéré et al. (2018)) is available at https://www.icos-cp.eu/science-and-impact/global-carbon-budget/2018. The two atmospheric inversion datasets are available in Bastos et al. (2019). The monthly sea level pressure is from ERA5 reanalysis available at Climate data store, for the period 1959-1978 is from Bell et al. (2020) and the period 1979-2017 is from Hersbach et al. (2019). The download links for the 15 teleconnection indices are available in Table 1 of Section 2.1.2, last access on August 2021. The sea level pressure and NBP from CESM1.2 is available in Stolpe et al. (2019).

## Appendix C: Code availability

The Python scripts are available at https://edmond.mpdl.mpg.de/privateurl.xhtml?token=8f717b4f-aea2-4a9b-96a3-efc8487e54af

*Author contributions.*

NL processed the data, performed the analysis and wrote the first draft of the manuscript. SS, AB, MR designed the study. SS pre-processed the CESM1.2 data. All authors contributed to design the methodology and to the writing of the manuscript.

*Competing interests.*

The authors declare that they have no conflict of interest.

*Disclaimer.*

Publisher's note: Copernicus Publications remains neutral with regard to jurisdictional claims in published maps and institutional affiliations.

*Acknowledgements.* Markus Reichstein and Alexander J. Winkler acknowledge support by the European Research Council (ERC) Synergy Grant "Understanding and Modelling the Earth System with Machine Learning (USMILE)" under the Horizon 2020 research and innovation programme (Grant agreement No. 855187).

**References**

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
