# Peer review of "Inter-annual global carbon cycle variations linked to atmospheric circulation variability"

_EGUsphere, 2022_

## Referee Comment (RC1)

Review of the paper by Li et al:

**Inter–annual global carbon cycle variations linked to atmospheric circulation variability**

**General Comments:**

Na Li and co-authors relate the inter-annual variability (IAV) of de-trended global observed atmospheric $CO_2$ growth rates and the modelled global land sink from 1959 to 2017 with spatio-temporal sea level pressure (SLP) anomaly fields. They use a regularised linear regression method (Ridge Regression, RR) combined with a statistical learning technique to predict the IAV of the observed and model-simulated global $CO_2$ growth rates. They compare these results with a similar regression that is based on 15 classical global and hemispheric teleconnection indices, as well as with a regression that is solely based on Southern Oscillation index (SOI). They find very good predictability (Pearson R > 0.7) with boreal winter SLP anomalies, that is comparable or even better than with classical teleconnection indices. They show that $CO_2$ IAV is most sensitive to tropical and southern hemisphere SLP anomalies (a finding, which was already observed by Bacastow in 1976 and attributed to the influence of ENSO on the land biosphere sink by Keeling et al. in 1995).

This is an interesting and careful analysis, with the results being well presented in the manuscript. However, I would have appreciated some more discussions of the results. For example, it would be nice to gain some direct insight, which land regions dominate the globally observed atmospheric $CO_2$ growth rates. The biosphere models obviously reproduce the IAV very well so that this information should be available from these models.

**Specific comments:**

**Abstract**

Line 6: Please add "global" in "…from the **global** de-trended …"
    "… and from different datasets …": Please be more specific which datasets have been evaluated.

**1. Introduction**

Line 23: "Quantifying and understanding the patterns of variability in the C-cycle and their drivers is crucial to better understand the drivers of C-cycle dynamics and better constrain future climate projections." I fully agree to this statement, however, in the current study solely the SLP anomaly is correlated with the $CO_2$ IAV, which, at least to my understanding, serves as a place-holder for the real drivers, which are e.g. temperature, water and radiation availability, for $CO_2$ exchange with the land biosphere (as correctly stated in line 39). Do the correlations presented here really help "process understanding of C-cycle dynamics"? This needs to be explained to the reader or, alternatively, such rather strong statements should be a bit de-emphasised throughout the manuscript.

Line 30: "(e.g., carbon uptakes by photosynthesis)" Isn't heterotrophic respiration even less well observable?

Line 72: Please add again "… **global** atmospheric $CO_2$ …"

Line 74: "We additionally compare results with…" which results?

Line 77: Please make sure that the reader understands this sentence correctly, i.e. that the latitudinal domains only refer to SLP, not to the biosphere land sink. See my general remark above.

**2. Data and methods**

**2.1 $CO_2$ data sets:**

As an "atmospheric observations person", I was a bit confused that not only the AGR but also the modelled land sinks etc. were named "CO2 data sets" (see my comment on lines 225ff below). Also, please have a look at Le Quéré et al. (2018) how the different components of the carbon budget listed in Eq. (1) shall be cited (see their Table 2).

Lines 135-136: What are the consequences that "dynamic vegetation" is not included?

**2.2 Data pre-treatment:**

Line 145: "grid points"? Do you mean "months"?

Line 146: "… LOESS as for the SLP fields." Do you mean "as for the $CO_2$ time series"? There is no mentioning of a smoothing of the SLP fields.

**2.4 Experimental design:**

Line 206: "… from 1 to 53 years". Do you mean "1 to 35 years".

Lines 221-222: Verb is missing in the last sentence.

**3. Results and discussion**

**3.1 Global IAV patterns:**

Lines 225-227: See my earlier comment on the confusion about "observed" $CO_2$ time series (sec. 2.1). It would be easier for the reader if only the AGR is called an observed $CO_2$ time series and the biosphere model based IAV records are called differently. In this manuscript I had a hard time to get used to the many different terms and abbreviations. A few more explanatory words here and there may help to digest the text.

Line 233: include "… LOO correlation of **SLP-**predicted and observed**/modelled** $CO_2$ time series …"

Figure 2: It is a bit confusing that the y-Axis title is called $\rho_{SLP}$. I guess simply $\rho$ would be correct.

Figure 2 caption Line 1: insert "… annual **measured and modelled** $CO_2$ time-series…"
    Line 4: insert "…de-trended **data** based ….predicted vs. observed **and modelled** $CO_2$ time …"
    Line 5: "Additionally …" Verb is missing in this sentence.

Lines 258-259: "2) $SL_{Resid}$ implicitly includes the variability from land use changes as well as ocean sink variations" Any idea which one contributed more?

Line 293: insert "…number **of** predictors …"

**3.2 Sensitivity to the SLP domains:**

Lines 299-300 and 304-306: If I read the heat maps in Fig. 4 correctly, the predictability is largest if the domain includes high latitudes of the SH, i.e. not only the tropics.

Lines 311-315: This explanation would be more convincing with some spatial information on the biosphere fluxes (see my general comment).

Lines 316-317: "… is likely due to strong …" here a more detailed inspection of the model results may give insight (see my general comment).

**3.3 Sensitivity to the temporal domains:**

Lines 345-346 and Fig. 6: When increasing the time interval there are less possibilities to obtain different $\rho_{SLP}$ and the correlated data become more and more similar. Doesn't this automatically decrease the variability of $\rho_{SLP}$?

Lines 360 and 364: Perhaps better use the word "interval" instead of "scale".

An explanation of Figure 6b is missing in the text.

Line 395: please include "… different **atmospheric** driving …"

Lines 392-396: Please refer here to my comment that SLP is only a place-holder for atmospheric drivers influencing the C-cycle.

Figure A1: The x-axis scale and title should be degrees.

Figure A3: What are the light blue shaded areas?

Figure A6 caption line 2: delete "extending" at the end of the line.

---

## Referee Comment (RC2)

**Review of the manuscript egusphere-2022-96 : "Inter-annual global carbon cycle variations linked to atmospheric circulation variability" by Na Li, Sebastian Sippel, Alexander J. Winkler, Miguel D. Mahecha, Markus Reichstein and Ana Bastos**

The authors are investigating the ability of SLP anomaly field to predict global carbon inter-annual variability (IAV) when used in a ridge regression (RR). In particular, the IAV of de-trended global observed atmospheric $CO_2$ growth rates and modelled global land sink are reconstructed. This RR is compared to a another RR taking 15 teleconnection indices as predictors and to a linear regression only based on SOI. The use of SLP allows a good reconstruction of the different carbon cycle time-series IAV.

In general, the article is a bit difficult to follow. Indeed, the word 'global' is mentioned several times throughout the paper but its meaning is different whether it is $CO_2$ (single global value) or SLP (800 grid-point). An effort should be made to ease the reading. This paper is showing some potential. However, the paper needs some clarification/modification before publication.

Major comments :

A About the estimation procedure : what is the influence of the LOO consisting in using three consecutive years as test sample ? What would happen if the test sample is bigger ?

B About the SLP anomaly fields as predictors : predictor number evolve from 4 to 800 depending the predictors domain. How ever it seems impractical to perform multiple RR with up to 800 predictors to estimate one global value and select the best predictor domain. If the intend of the authors is to provide an alternative to study the relationship between C-cycle and circulation variability this can be perceived as heavy. Besides, based on Figure 2, the SLP-based RR is not necessarily better than the indices-based RR or the SOI-based linear regression. A user would be tempted to use one of those.

(a) The main problem is to compare results of regression with very different number of predictors only based on $\rho_{SLP}$. What is the trade-off between adding predictors and the RR improvement ? Since the objective is to capture the IAV, using the principal mode of variability of SLP fields instead of the entire fields could remedy the aforementioned issue. For instance, the first EOFs of SLP fields can be used as predictors. The number of EOF can be chosen according the proportion of the variance captured by the EOFs.

(b) RR is adapted for large numbers of predictors. It would be interesting to see the performances of a usual generalised linear model based on the EOFs of SLP fields.

Minor comments :

— line 29 : 'plagued' may be a little harsh

— line 42 : Replace 'These dynamics' by 'These climate variability modes'. These variability modes may be subject to irreducible noise but they can not be considered as "noise", please rephrase this.

— line 68 : In "while at the same time", at the same is redundant.

— Section Data pre-treatment : clarify this section as follows : 1) trend removing (CO2, SLP and indices) and anomalies computing (SLP) 2) spatial and temporal aggregation.

— from line 300 : scale is used to refer to the spatial predictor domain or temporal learning periods. Please be precise, in those case scale is not appropriate.

— line 328 : Maybe 2001 instead 2003 ?

---

## Author Comment (AC1)

**Reply to Reviewers for the manuscript egusphere-2022-96:" Inter–annual global carbon cycle variations linked to atmospheric circulation variability":**

Na Li, Sebastian Sippel, Alexander Winkler, Miguel D. Mahecha, Markus Reichstein, Ana Bastos

**Reply to Reviewer #1**

Na Li and co-authors relate the inter-annual variability (IAV) of de-trended global observed atmospheric CO2 growth rates and the modelled global land sink from 1959 to 2017 with spatio- temporal sea level pressure (SLP) anomaly fields. They use a regularised linear regression method (Ridge Regression, RR) combined with a statistical learning technique to predict the IAV of the observed and model-simulated global CO2 growth rates. They compare these results with a similar regression that is based on 15 classical global and hemispheric teleconnection indices, as well as with a regression that is solely based on Southern Oscillation index (SOI). They find very good predictability (Pearson R > 0.7) with boreal winter SLP anomalies, that is comparable or even better than with classical teleconnection indices. They show that CO2 IAV is most sensitive to tropical and southern hemisphere SLP anomalies (a finding, which was already observed by Bacastow in 1976 and attributed to the influence of ENSO on the land biosphere sink by Keeling et al. in 1995).

We thank the reviewer for the positive evaluation of our study and the constructive comments. We provide in-depth replies to each comment below.

This is an interesting and careful analysis, with the results being well presented in the manuscript. However, I would have appreciated some more discussions of the results. For example, it would be nice to gain some direct insight, which land regions dominate the globally observed atmospheric CO2 growth rates. The biosphere models obviously reproduce the IAV very well so that this information should be available from these models.

We thank the reviewer for this suggestion to improve the manuscript. In the revised version of the manuscript, we now provide a more in-depth discussion about processes and how this study can contribute to improved understanding of IAV in the carbon cycle.

**Specific comments:**

**Abstract**

Line 6: Please add "global" in "...from the **global** de-trended ...", "... and from different datasets ...": Please be more specific which datasets have been evaluated.

Thanks, we corrected this phrase in line 6, which now reads: "CO2 variability is diagnosed from the **global** detrended atmospheric CO2 growth rate and the land CO2 sink from **16 dynamic global vegetation models and two atmospheric inversions** different datasets in the global carbon budget **2018**."

**1. Introduction**

Line 23: "Quantifying and understanding the patterns of variability in the C-cycle and their drivers is crucial to better understand the drivers of C-cycle dynamics and better constrain future climate projections." I fully agree to this statement, however, in the current study solely the SLP anomaly is correlated with the CO2 IAV, which, at least to my understanding, serves as a place-holder for the real drivers, which are e.g. temperature, water and radiation availability, for CO2 exchange with the land biosphere (as correctly stated in line 39). Do the correlations presented here really help "process understanding of C-cycle dynamics"? This needs to be explained to the reader or, alternatively, such rather strong statements should be a bit deemphasised throughout the manuscript.

We thank the reviewer for pointing this out, and we agree that the relevance for process understanding was overemphasized. Still, we believe that our results have important implications for the analysis of drivers of variability of C–cycle processes, given the short nature of observational records compared to the time–scales of natural climate variability modes. We explain our reasoning below and have updated the manuscript accordingly.

Indeed, temperature, water, and radiation availability, etc. are the direct drivers of ecosystem processes that control C fluxes (photosynthesis, growth, decomposition, fires, ...). These drivers, however, show strong covariations in time and space, for example due to land–atmosphere feedbacks (Seneviratne et al., 2012), but also because they are influenced to a large extent by atmospheric circulation patterns that affect anomalies in multiple variables at the same time, leading to spatio–temporal co–variability. For example, persistent anticyclonic conditions promote both warm, sunny, and dry conditions at the surface in summer. Likewise, ENSO variability controls anomalies in both temperature and water/radiation availability over the tropics and in the extratropics through large–scale teleconnections. As already observed by Bacastow (1976), ENSO explains a large fraction of variability in the global carbon cycle. It has been argued that indices reflecting atmospheric circulation patterns might be more useful predictors of variability in ecosystem activity than the direct drivers themselves because these indices aggregate information about the range of climatic conditions experienced by ecosystems at a particular time and place, so that they can be used as a way to reduce dimensionality of the space of climatic drivers (Hallet et al., 2004; Bastos et al., 2016; 2017; Zhu et al., 2017).

Here we took a step forward and adopted the approach proposed by Sippel et al., (2019), where SLP anomaly fields are used directly as a proxy of atmospheric circulation variability. The advantage of this approach is that it allows for a more flexible definition of the relevant atmospheric circulation patterns that influence interannual CO2 variability, since it does not require the use of predefined teleconnection indices. This allows, for example, identifying relevant domains that are affected by more than one mode of atmospheric variability, such as the west Pacific domain in MAM (Fig. 3), or by variations in the importance of different atmospheric domains over time (Fig. 4). Nevertheless, we agree that for interpretation of the correlations found, it is important to understand how the identified SLP patterns influence the direct climatic drivers of CO2 sinks and sources. For this, we evaluated the Pearson correlations between global SLP predicted AGRR (i.e. the component of AGRR that is driven by the atmospheric circulation patterns in Fig. 3) to global land temperature and precipitation anomaly, both from CRU\_TS4.05 monthly data, over the period 1980–2017. The corresponding maps are shown in Fig. R1:

---

## Author Comment (AC2)

**Reply to Reviewers for the manuscript egusphere-2022-96:" Inter–annual global carbon cycle variations linked to atmospheric circulation variability":**

**Na Li, Sebastian Sippel, Alexander Winkler, Miguel D. Mahecha, Markus Reichstein, Ana Bastos**

**Reply to Reviewer #2**

The authors are investigating the ability of SLP anomaly field to predict global carbon inter-annual variability (IAV) when used in a ridge regression (RR). In particular, the IAV of de-trended global observed atmospheric $CO_2$ growth rates and modelled global land sink are reconstructed. This RR is compared to a another RR taking 15 teleconnection indices as predictors and to a linear regression only based on SOI. The use of SLP allows a good reconstruction of the different carbon cycle time-series IAV.

In general, the article is a bit difficult to follow. Indeed, the word 'global' is mentioned several times throughout the paper but its meaning is different whether it is $CO_2$ (single global value) or SLP (800 grid-point). An effort should be made to ease the reading. This paper is showing some potential. However, the paper needs some clarification/modification before publication.

Thanks for the in–depth comments and constructive suggestions. We address each point separately below.

Major comments:

A About the estimation procedure: what is the influence of the LOO consisting in using three consecutive years as test sample? What would happen if the test sample is bigger?

We would like to thank the reviewer for pointing this out. Since we have only limited samples (less than 60), we selected leave–one–out (LOO) rather than other cross–validation approaches with bigger test samples. We agree though that results may be sensitive to this choice. We select three consecutive years, but only the year in the center is used as a test sample each time. We remove the years before and after the test sample to avoid the influence of time–series autocorrelations. We note that in the dependence study (Fig. A3 of the manuscript), the time–series autocorrelation in $CO_2$ IAV is relatively small, so that this step might not have a strong influence on the predictability. However, the predictability also depends on the stationarity (particularly variability) of the dataset: the more stable the variance is, the less predictability difference when using leave–one–out or k–fold. Here, if the test sample is bigger, varying predictability is expected. To verify this, we conducted the k–fold cross–validation, with test sample sizes of k=1, 2, 3, 6, and 9. This time, all selected test data are predicted (our LOO

approach removed the samples with year before and after the middle year), and we run 100 times of each estimation with different shuffled train/test grouping. Below we show the resulting correlations between predicted and test values:

[Figure]

Figure R7. Comparison of global seasonal SLP predictability with different estimation approaches: LOO of our approach (leave–one–out with the samples before and after the test year removed from test and train group), leave–one–out, and leave 2, 3, 6, 9 out approaches, in the period 1959–2017.

The predictability shows the larger spread, the larger the number of test samples is, but the median remains relatively stable. Note that the k=1 leave–one–out approach has no uncertainty in this experiment, since no matter how many times we shuffle the train/test dataset, the predictability is the same. There is some difference between the LOO and k=1 cross–validation, it might be due to the removal of the years before and after the test sample. In such a case, the LOO has two less train samples (same train samples as k=3) compared to k=1 cross–validation, which might influence the predictability due to different train sample numbers. This test run shows that the smaller the test sample sizes, the more robust the predictability. We have added one note on this in the discussion text (line 168–169), which now reads:

" Given the relatively short period (n < 60), **and generally the smaller the number of the test samples, the more robust the predictability,** here we use leave–one–out (LOO)… "

B About the SLP anomaly fields as predictors: predictor numbers evolve from 4 to 800 depending the predictors domain. How ever it seems impractical to perform multiple RR with up to 800 predictors to estimate one global value and select the best predictor domain. If the intend of the authors is to provide an alternative to study the relationship between C-cycle and circulation variability this can be perceived as heavy. Besides, based on Figure 2, the SLP-based RR is not necessarily better than the indices-based RR or the SOI-based linear regression. A user would be tempted to use one of those.

We agree with the reviewer that "the global SLP–based RR is not necessarily better than the indices–based RR" according to Fig. 2 in the manuscript. If the user wants to evaluate a simple correlation between C–cycle and atmospheric circulation variability, teleconnection indices may yield equally good predictions. However, this approach might be too inflexible for certain applications, where identifying more general patterns of atmospheric circulation driving variability in the target variable might provide more comprehensive information. This is likely the case for local/regional studies for climate variables such as precipitation or temperature, as shown in Sippel et al. (2020), where the SLP–based RR was shown to be a more robust approach than using EOF–based circulation components for regression.

A more fundamental justification for this approach is that teleconnection indices summarize the variability of different modes of atmospheric circulation in simple time–series, where the spatial patterns of SLP, SST or associated variables for calculating the indices are usually fixed. This can lead to multiple, slightly different definitions of the same mode, for example the multiple indices that can be used to describe the El–Niño/Southern–Oscillation phenomenon. Moreover, teleconnections are known to interact, so that the resulting circulation pattern is a combination of different modes, for example, ENSO, SAM and IOD together influence the Australian precipitation and drought (Cleverly et al., 2016). Finally, teleconnection indices reflect the dominant modes of atmospheric variability, which are not necessarily the dominant atmospheric circulation patterns controlling $CO_2$ variability.

Our approach could identify the spatial patterns of SLP variability driving most of global $CO_2$ IAV (Fig. 3a in the manuscript) these patterns include the ENSO pattern but also reveal other important regions which do not correspond necessarily to a single index (e.g. the west Pacific area in MAM, which can be associated with multiple teleconnection patterns as discussed in the reply to Reviewer #1).

(a) The main problem is to compare results of regression with very different number of predictors only based on $\rho_{SLP}$. What is the trade-off between adding predictors and the RR improvement? Since the objective is to capture the IAV, using the principal mode of variability of SLP fields instead of the entire fields could remedy the aforementioned issue. For instance, the first EOFs of SLP fields can be used as predictors. The number of EOF can be chosen according the proportion of the variance captured by the EOFs. (b) RR is adapted for large numbers of predictors. It would be interesting to see the performances of a usual generalised linear model based on the EOFs of SLP fields.

We thank the reviewer for pointing out this critical and important issue. The Ridge Regression is especially tailored to address the problem of a large number of (correlated) predictors, since it attributes low weights to predictors that carry little additional information, while keeping those that do contribute the most to the variance of the target variable. Furthermore, we performed a preliminary sensitivity study as stated in section 2.4.1. By using different resolutions of SLP at

2 ° * 2 ° (16200 predictors), 5 ° * 5 ° (4536), and 9 ° * 9 ° (800) (Fig. A1 in the manuscript). The predicted correlations show only a slight difference with the changing number of predictors (e.g., r varied less than 0.03 with $AGR_R$), as expected by the use of RR.

The reason we did not use EOF of SLP fields is that EOF would capture the main variance of the SLP field, but not necessarily that of the main patterns that influence $CO_2$ IAV (similar to our reasoning about teleconnection indices in the reply above). Moreover, mathematically Ridge regression and EOF regression are deeply connected: EOF regression cuts off all components with small variance beyond a certain threshold, while ridge regression shrinks them, which allows that some information that is hidden in low variance components can still be used for prediction (von Wieringen et al., 2021). This approach might reveal hidden components of SLP that are driving $CO_2$ IAV. A comparison of these two approaches has been performed in Sippel et al., (2019), where the performance of RR and EOF when using SLP anomalies to predict temperature/precipitation variations are compared. They show that RR performs better than EOF (hence also justifying our answer to the previous comment). Finally, we followed the reviewer's advice and compared our results with those based on a generalized linear model using the first EOFs of SLP fields. The results are shown in Fig. R8.

First, we selected the first 10 components of SLP anomalies (DJF) using EOF analysis. We did this for three different SLP spatial domains (global, 18° N-18° S, and 18° N-72° S), and for the periods 1959-2017 and 1980-2017 separately.

We then reconstruct SLP fields based on an increasing number of components, from 1 (based on components 1), explaining 25% in period 1959-2017 and 23% in period 1980-2017 of the variance in SLP, to 10 components (1… 10), explaining 75% and 79% of the variance in SLP. We use these as predictors of $AGR_R$ in a simple linear regression (also with LOO estimation).

The results are then compared with those using RR (Fig. R8)

[Figure]

Figure R8. Comparison of EOF–linear to RR approach under different DJF SLP spatial domains. The upper plots are the comparison of predictability, The y label shows the Pearson correlation between the predicted $CO_2$ IAV to original $CO_2$ IAV. The solid line represents the results by using the EOF–linear approach under a different number of extracted SLP EOF components that are included in the linear regression. The dashed line represents the results by using RR. Different colored lines represent different SLP domains used. The lower plots are the corresponding explained SLP variance by different EOF components.

The EOF–linear approach returns in most cases lower predictability than RR, depending on how many components are included. According to Fig. 4 in the manuscript, the domain from 18° N to 72° S in DJF shows the highest predictability (r=0.81 in 1980–2017 and r=0.73 in 1959–2017) when using RR. In this SLP domain, the EOF-linear approach generally shows much lower

predictability with all different numbers (1~10) of components included (r < 0.75 in 1980–2017 and r < 0.7 in 1959-2017).

This test shows that a EOF–linear approach can generally achieve lower/similar predictability than RR when selecting global/tropical SLP anomalies, which is consistent with the mathematical explanation above.

Minor comments :

— line 29: 'plagued' may be a little harsh

We thank the reviewer for pointing this out, this sentence has been corrected to: "since some of these processes are  confounded by large uncertainties".

— line 42 : Replace 'These dynamics' by 'These climate variability modes'. These variability modes may be subject to irreducible noise but they can not be considered as "noise", please rephrase this.

Thanks, the sentence has been corrected to:" These  **climate variability modes** are generated within the coupled atmosphere–ocean …"

— line 68 : In "while at the same time", at the same is redundant.

We agree with the reviewer and have removed "at the same time".

— Section Data pre-treatment : clarify this section as follows : 1) trend removing (CO2, SLP and indices) and anomalies computing (SLP) 2) spatial and temporal aggregation.

We thank the reviewer for the good advice. We now add paragraph headers for "Trend removal" and "Spatial and temporal aggregation". We put the pre–treatment of $CO_2$ and teleconnection indices under "trend removal" and SLP under "spatial and temporal aggregation". We have made some changes in the treatment of teleconnection indices. The new text in section 2.2 Data pre–treatment is:

***Trend removal***

The long–term trend of $CO_2$ time–series was removed by locally weighted scatterplot smoothing (LOWESS) of the annual time–series with fixed window size of 25 % interval longer than 30 years (1959–2017) and 45 % for shorter period (1980–2017). For monthly teleconnection indices, we first  calculate DJF, MAM, JJA, and SON mean values , **we then remove the long–term trends by applying the LOWESS as for the $CO_2$ time–series** , and further include DJF and MAM combined (DJF+MAM) as treated in SLP (as described below).

*Spatial and temporal aggregation*

The monthly mean SLP fields are area–weighted and aggregated to 2 ° * 2 °, 5 ° * 5 °, and 9 ° * 9 ° spatial resolution, and the seasonal cycle removed by subtracting the monthly mean values for each pixel. We then aggregate SLP values in seasonal means for: December of the previous year to February of each given year (DJF), March–May (MAM), June–August (JJA), and September–November (SON) and further consider DJF and MAM combined (DJF+MAM), so the number of **pixel-based time–series (predictors)**  in DJF+MAM is double of DJF. **Note that a large fraction of the pixel–based time–series of seasonal SLP anomalies show no long–term trend, and the predicted differences between LOWESS detrended and no detrended SLP are small. Here we keep the analysis of SLP anomaly with no LOWESS detrending.** Here, we refer to DJF and MAM as boreal winter and boreal spring."

We would like to note that the SLP fields were not detrended (see reply to Reviewer #1). We have added two sentences in the above text:

**"Note that a large fraction of the pixel-based time–series of seasonal SLP anomalies show no long–term trend, and the predicted differences between LOWESS detrended and no detrended SLP are small. Here we keep the analysis of SLP anomaly with no LOWESS detrending."**

— from line 300 : scale is used to refer to the spatial predictor domain or temporal learning periods. Please be precise, in those case scale is not appropriate.

We thank the reviewer for pointing out, "scale of" has been removed.

— line 328 : Maybe 2001 instead 2003 ?

We thank the reviewer for correcting that, it is "2001" and has been corrected.

---

## Author Response (AR1)

**Reply to Reviewers for the manuscript egusphere-2022-96:" Inter–annual global carbon cycle variations linked to atmospheric circulation variability":**

**Na Li, Sebastian Sippel, Alexander Winkler, Miguel D. Mahecha, Markus Reichstein, Ana Bastos**

**Reply to Reviewer #1 and #2**

We thank the reviewers for the concrete comments. Here we address the manuscript according to the comments. In addition, based on the comments, we made some extra changes in the manuscript for easier reading and understanding.

The main extra changes including:

(1) We added "global" before "C-cycle IAV", unless referring to regional C–cycle IAV.

(2) To be consistent, we changed the relevant terms such as "atmospheric variability", "climatic variability", … consistently to "large–scale atmospheric circulation variability/modes".

(3) The "predictability" means how well the $CO_2$ time–series can be predicted. The "predictive skill" means how well SLP/teleconnection indices as predictors in RR, to predict/explain $CO_2$ time–series. Here we use more "predictability" or "predictive skill" instead of using "rSLP" or "rTele" for better understanding.

(4) For a better understanding, we have added "Ridge Regression" before "coefficients" in most cases. In many places, we use "Ridge Regression coefficients" instead of "wSLP" or "wTele".

(5) We try to avoid too many abbreviations, so "NH", "SH", "RR", "GCB2018", and "LOO" have all been spelled out as "Northern Hemisphere", "Southern Hemisphere", "Ridge Regression", "Global Carbon Budget 2018", and "leave–one–out".

(6) "DJF" and "MAM" in some places are changed to "winter" and "spring".

(7) Other grammar and small errors correction, missing citations, or text changes for easier reading and understanding.

(8) Figure A1 has been updated from line plots to point plots.

(9) Figure 4, Figure A7, and Figure A8 have been updated. For Figure 4, we changed the x-axis of the top right plot. Before, the corresponding x-axis of this plot was at the bottom right, now we moved it to the middle of Figure 4 and directly under the top right plot. Figure A7 and A8 also have the x-axis moved up to middle.

Below we address the changes according to the comments **(the line numbers in replies are according to the new manuscript that showing changes)**. Note that some changes in the manuscript are slightly different with the first version of "reply to the reviewer".

**Changes according to the reviewer #1:**

**Specific comments:**

**Abstract**

Line 6: Please add "global" in "...from the **global** de-trended ...", "... and from different datasets ...": Please be more specific which datasets have been evaluated.

Line 6-8 (below all the line numbers in the replies are according to the new manuscript with changes). Thanks, we corrected this phrase in line 6, which now reads: "**C-cycle**  variability is diagnosed from the **global** detrended atmospheric $CO_2$ growth rate and the land $CO_2$ sink from  **16 dynamic global vegetation models and two atmospheric inversions**  in the Global Carbon Budget **2018**."

**1. Introduction**

Line 23: "Quantifying and understanding the patterns of variability in the C-cycle and their drivers is crucial to better understand the drivers of C-cycle dynamics and better constrain future climate projections." I fully agree to this statement, however, in the current study solely the SLP anomaly is correlated with the CO2 IAV, which, at least to my understanding, serves as a place-holder for the real drivers, which are e.g. temperature, water and radiation availability, for CO2 exchange with the land biosphere (as correctly stated in line 39). Do the correlations presented here really help "process understanding of C-cycle dynamics"? This needs to be explained to the reader or, alternatively, such rather strong statements should be a bit de-emphasised throughout the manuscript.

We have added two supplementary Figures in appendix (Fig. A11 and Fig. A12) in the manuscript, also added the below text to line 301-319 in the manuscript:

"In  **winter the positive** coefficients over the eastern tropical Pacific are higher than in other regions, , which are influenced by El Niño and La Niña respectively (Monahan, 2001; Hsieh, 2004; Rodgers et al., 2004; Schopf and Burgman, 2006; Sun and Yu, 2009; Yu and Kim, 2011): El Niño induces negative SLP anomalies over the East Pacific and positive SLP anomalies over the west Pacific (see (King et al. (2020), Fig. 5).  **The results are consistent with** the land sink  **being** negatively driven by  ENSO in winter**:** strong El Niño, decreased land sink**,**  **and**  **winter** strong La Niña, increased land sink. **In spring, the area over the central and western tropical Pacific shows stronger coefficients, and likely corresponds to a mix of different modes, such as the ENSO, West Pacific teleconnection and the Interdecadal Pacific Oscillation, all showing strong coefficients in Fig. 3b (SOI, WP and TPI indices). In Fig. A11 we show the anomalies in temperature and precipitation associated to these patterns, as well as those in NBP from the two atmospheric inversions (Fig. A12). Generally, the temperature anomalies over the tropics show negative correlations to annual land sink (SLP driven AGR$_R$) in both winter (as high as -0.85) and spring (as high as -0.73), while weaker but positive correlations**

**are found in Eurasia. Tropical precipitation anomalies show roughly positive correlations in winter (as high as 0.73) and in spring (as high as 0.67). This pattern indicates that AGR$_R$ is generally higher for cooler and wetter conditions over the tropics and Southern Hemisphere semi-arid regions in both seasons, which result in increased NBP (Fig. A12), as well as cooler but predominantly drier conditions over Eurasia, which result in a complex pattern of NBP anomalies (Fig. A12). These results are consistent with the strong ENSO fingerprint on the IAV of global $CO_2$ atmospheric growth rate and global land sink, e.g. as pointed out by Piao et al. (2020) and with the importance of southern semi-arid ecosystems (Ahlström et al., 2015), for IAV in the global land sink."**

Note that the Fig A11 (spatial distribution of correlations to temperature and precipitation) is slightly different than the Fig. R1 in the first version of "reply to reviewer". Before, the spatial correlation distribution is plotted based on temperature and precipitation both with resolution of 4.5 ° * 4.5°. But the ocean is not nicely masked out and the figures look bit rough. So we updated the two figures with temperature and precipitation both with the resolution of 1 ° * 1 °, also the contour level when plotting has increased. The new correlations are a bit different: "Generally, the temperature anomaly over the tropics shows negative correlation to land sink (SLP driven AGRR) in both DJF (as high as −0.85) and MAM (as high as  −0.73)…", "Tropical precipitation anomaly shows roughly positive correlation in DJF (as high as  0.73) and in MAM (as high as  0.67)." But the spatial patterns remain similar.

Line 30: "(e.g., carbon uptake by photosynthesis)" Isn't heterotrophic respiration even less well observable?

Line 34-35: We agree with the reviewer that "heterotrophic respiration is even less well observable", we have changed it to "(e.g. photosynthesis **or heterotrophic respiration**) (Schimel et al., 2015; **Basile et al., 2020**)".

Line 72: Please add again "... **global** atmospheric CO2 ..."

Line 78: Thanks, added: "We use observation–based time–series of **global** atmospheric $CO_2$ growth rate…"

Line 74: "We additionally compare results with..." which results?

Line 81, 86, We thank the reviewer for pointing this out, we deleted the sentence: "", and we explained this in line 86, "**by comparing the fraction of C–cycle IAV that can be explained by large atmospheric circulation variability based on these datasets with that of a very long time–series (4000 years) of land CO2 fluxes simulated by the Community Earth System Model (CESM).**"

Line 77: Please make sure that the reader understands this sentence correctly, i.e. that the latitudinal domains only refer to SLP, not to the biosphere land sink. See my general remark above.

Line 84-85, We thank the reviewer for pointing out this aspect, modified and specified as: "Next, we analyze and discuss how the **global** C–cycle sensitivity to atmospheric circulation  **variability** from various latitudinal domains of **SLP anomaly fields** (Section 3.2)."

**2. Data and methods 2.1 CO2 data sets:**

As an "atmospheric observations person", I was a bit confused that not only the AGR but also the modelled land sinks etc. were named "CO2 data sets" (see my comment on lines 225ff below). Also, please have a look at Le Quéré et al. (2018) how the different components of the carbon budget listed in Eq. (1) shall be cited (see their Table 2).

Line 95, Thanks for pointing out, the land sinks have been referred to as "modelled" in the revised version of the manuscript. The citations for Equation 1 have been added: "…emissions from fossil fuel (FF) **(Boden et al., 2017; UNFCCC, 2018; Peters et al., 2011b)** and land use change (FLUC) **(Houghton and Nassikas, 2017; Hansis et al., 2015)**, The AGR **(Dlugokencky and Tans, 2018)**, the carbon uptake by the ocean sink (SO) and the land sink (SL) **(references for individual models of SO and SL can be found in Table 4 of Le Quéré et al. (2018))**."

Lines 135-136: What are the consequences that "dynamic vegetation" is not included?

No suggested changes in the comment.

**2.2 Data pre-treatment:**

Line 145: "grid points"? Do you mean "months"?

Line 166-167, we thank the reviewer for pointing this out. Here the "grid points" refers to the number of pixel-based SLP time series (predictors) selected, corrected in the manuscript as: "so the number of  **pixel–based time–series (predictors)** in DJF+MAM is double of DJF."

Line 146: "... LOESS as for the SLP fields." Do you mean "as for the CO2 time series"? There is no mentioning of a smoothing of the SLP fields.

Since the differences between SLP detrending/non detrending are small, and given the reasoning explained above, we keep the analysis of SLP anomaly with no LOWESS detrending. However, we add two sentences in the section 2.2 Data pre-treatment to indicate the possible influences of SLP trends:

Line 168, "**Note that a large fraction of the pixel–based time–series of seasonal SLP anomalies show no long–term trend, and the predicted differences between LOWESS detrended and not detrended SLP are small. Here we keep the analysis of SLP anomalies with no LOWESS detrending.**"

 **2.4 Experimental design:**

Line 206: "... from 1 to 53 years". Do you mean "1 to 35 years".

Line 233, Corrected to: "the temporal auto–correlation of all $CO_2$ time–series is mostly less than 0.4 with lag ranging from 1 to  **35** years"

Lines 221-222: Verb is missing in the last sentence.

Line 248-249, thanks, the sentence has now been corrected:

"The error rate is calculated by the number of invalid predictions that  **have** significance P > 0.05  divided by the number of total predictions within a given window."

**3. Results and discussion**

**3.1 Global IAV patterns:**

Lines 225-227: See my earlier comment on the confusion about "observed" CO2 time series (sec. 2.1). It would be easier for the reader if only the AGR is called an observed CO2 time series and the biosphere model based IAV records are called differently. In this manuscript I had a hard time to get

used to the many different terms and abbreviations. A few more explanatory words here and there may help to digest the text.

Line 233: include "... LOO correlation of **SLP-**predicted and observed**/modelled** CO2 time series ..."

Line 267, We agree with the reviewer and have removed this paragraph and rephrased in the first paragraph of this section. We have included "/modeled" in other places as "observed/**modeled** CO2 time–series". We also specified in other places the "predictive skill of SLP", or "by using SLP as predictors" to specify the SLP–predicted results.

Figure 2: It is a bit confusing that the y-Axis title is called rSLP. I guess simply r would be correct.

"SLP" is now removed.

Figure 2 caption Line 1: insert "... annual **measured and modelled** CO2 time-series..."

For consistence with the addition above, we correct to: "Standardized annual **observed/modeled** CO$_2$ time–series over period 1959–2017"

Line 4: insert "...de-trended **data** based ...predicted vs. observed **and modelled** CO2 time ..." Line 5: "Additionally ..." Verb is missing in this sentence.

Added, thanks. The corrected lines: "in period 1980–2017 are detrended **data** based on their relevant period, and compared with detrended **data** based on 1959–2017, the difference is negligible. (b)  Pearson correlation of predicted vs observed/**modeled** CO$_2$ time–series based on the  **Ridge Regression** with SLP fields**…",** and "Additionally, Pearson correlation of predicted vs observed/modeled CO$_2$ time–series by linear regression **is** based on the single predictor of SOI index."

Lines 258-259: "2) SLResid implicitly includes the variability from land use changes as well as ocean sink variations" Any idea which one contributed more?

No suggested changes in the comment.

Line 293: insert "...number **of** predictors ..."

Line 348, added, thanks. The corrected line: "and the large number **of** predictors for  **Ridge Regression** training…"

**3.2 Sensitivity to the SLP domains:**

Lines 299-300 and 304-306: If I read the heat maps in Fig. 4 correctly, the predictability is largest if the domain includes high latitudes of the SH, i.e. not only the tropics.

Line 3567-358, We agree with the reviewer and have added the "high latitudes of the Southern Hemisphere". The corrected line: "We find improved predictability in both seasons when selecting smaller **spatial** domains (particularly  **including the tropics to high latitudes of the Southern Hemisphere**)".

Lines 311-315: This explanation would be more convincing with some spatial information on the biosphere fluxes (see my general comment).

No suggested changes in the comment.

Lines 316-317: "... is likely due to strong ..." here a more detailed inspection of the model results may give insight (see my general comment).

No suggested changes in the comment.

**3.3 Sensitivity to the temporal domains:**

Lines 345-346 and Fig. 6: When increasing the time interval there are less possibilities to obtain different rSLP and the correlated data become more and more similar. Doesn't this automatically decrease the variability of rSLP?

No suggested changes in the comment.

Lines 360 and 364: Perhaps better use the word "interval" instead of "scale".

Line 434, We thank the reviewer for pointing this out, we have changed accordingly: "We find that with different time  intervals…".

An explanation of Figure 6b is missing in the text.

We apologize, we realize the figure shows redundant information to that in panel (a) and have therefore deleted the panel (b) and the relevant description in the caption.

Line 395: please include "... different **atmospheric** driving ..."

Line 478, we have corrected the sentence to:

"**This method allows quantifying the contribution of atmospheric dynamical processes in driving variability in the C-cycle at global and regional scales, which may further be useful for attributing observed changes to internal climate variability versus anthropogenic climate change.**"

Lines 392-396: Please refer here to my comment that SLP is only a place-holder for atmospheric drivers influencing the C-cycle.

No suggested changes in the comment.

Figure A1: The x-axis scale and title should be degrees.

We thank the reviewer for pointing out this critical error, it is now corrected.

Figure A3: What are the light blue shaded areas?

The shaded areas are the 95% confidence interval of the calculated autocorrelation under different lags. We have now added this information in the Fig. A3 caption: "**The shaded areas are the 95% confidence interval of the calculated autocorrelation under different lags**".

Figure A6 caption line 2: delete "extending" at the end of the line.

Thanks for pointing this out, "extending" is removed.

**Changes according to the reviewer #2:**

**Major comments:**

A About the estimation procedure: what is the influence of the LOO consisting in using three consecutive years as test sample? What would happen if the test sample is bigger?

In our first reply, we have added one note on this in the discussion text (line 193–194), which now reads:

" Given the relatively short period (n < 60), **and generally the smaller the number of the test samples, the more robust the predictability,** here we use leave–one–out (LOO)... "

But now we consider this sentence might be too strong, so we decided not to add this sentence in the manuscript. But this sentence still holds true in this study.

B About the SLP anomaly fields as predictors: predictor numbers evolve from 4 to 800 depending the predictors domain. However it seems impractical to perform multiple RR with up to 800 predictors to estimate one global value and select the best predictor domain. If the intend of the authors is to provide an alternative to study the relationship between C-cycle and circulation variability this can be perceived as heavy. Besides, based on Figure 2, the SLP-based RR is not necessarily better than the indices-based RR or the SOI-based linear regression. A user would be tempted to use one of those.

No suggested changes in the comment.

(a) The main problem is to compare results of regression with very different number of predictors only based on ρSLP . What is the trade-off between adding predictors and the RR improvement? Since the objective is to capture the IAV, using the principal mode of variability of SLP fields instead of the entire fields could remedy the aforementioned issue. For instance, the first EOFs of SLP fields can be used as predictors. The number of EOF can be chosen according the proportion of the variance captured by the EOFs. (b) RR is adapted for large numbers of predictors. It would be interesting to see the performances of a usual generalised linear model based on the EOFs of SLP fields.

No suggested changes in the comment.

Minor comments :

— line 29: 'plagued' may be a little harsh

Line 32, we thank the reviewer for pointing this out, this sentence has been corrected to: "**Separating these effects is difficult because of the large uncertainties associated with some processes**..."

— line 42 : Replace 'These dynamics' by 'These climate variability modes'. These variability modes may be subject to irreducible noise but they can not be considered as "noise", please rephrase this.

Line 46, thanks, the sentence has been corrected to: " **These climate variability modes** are generated within the coupled atmosphere–ocean ..."

— line 68 : In "while at the same time", at the same is redundant.

Line 74, we agree with the reviewer and have removed "at the same time".

— Section Data pre-treatment : clarify this section as follows : 1) trend removing (CO2, SLP and indices) and anomalies computing (SLP) 2) spatial and temporal aggregation.

We thank the reviewer for the good advice. We now add paragraph headers for "Trend removal" and "Spatial and temporal aggregation". We put the pre–treatment of CO2 and teleconnection indices under "trend removal" and SLP under "spatial and temporal aggregation". We have made some changes in the treatment of teleconnection indices. The new text in section 2.2 Data pre–treatment is:

*Trend removal*

The long–term trend of $CO_2$ time–series is removed by locally weighted scatterplot smoothing (LOWESS) (Cleveland et al., 1991) of the annual time–series with fixed window size of 25 % interval longer than 30 years (1959–2017) and 45 % for shorter period (1980–2017). For monthly teleconnection indices, we first calculate the seasonal mean values of DJF, MAM, JJA, and SON, **we then remove the seasonal long–term trends by applying the LOWESS as for the $CO_2$ time–series**, and further include DJF and MAM combined (DJF+MAM) as treated in SLP (as described below). **(note that this paragraph is slightly different with the first version of "reply to reviewer")**

*Spatial and temporal aggregation*

The monthly mean SLP fields are area–weighted and aggregated to 2 ° * 2 °, 5 ° * 5 °, and 9 ° * 9 ° spatial resolution, and the seasonal cycle removed by subtracting the monthly mean values for each pixel. We then aggregate SLP values in seasonal means for: December of the previous year to February of each given year (DJF), March–May (MAM), June–August (JJA), and September– November (SON) and further consider DJF and MAM combined (DJF+MAM), so the number of **pixel-based time–series (predictors)**  in DJF+MAM is double of DJF.  **Note that a large fraction of the pixel–based time–series of seasonal SLP anomalies show no long–term trend, and the predicted differences between LOWESS detrended and no detrended SLP are small. Here we keep the analysis of SLP anomalies with no LOWESS detrending. Here,** we refer to DJF and MAM as boreal winter and boreal spring."

We would like to note that the SLP fields were not detrended (see reply to Reviewer #1). We have added two sentences in the above text:

**"Note that a large fraction of the pixel-based time–series of seasonal SLP anomalies show no long–term trend, and the predicted differences between LOWESS detrended and not detrended SLP are small. Here we keep the analysis of SLP anomalies with no LOWESS detrending."**

— from line 300 : scale is used to refer to the spatial predictor domain or temporal learning periods. Please be precise, in those case scale is not appropriate.

Line 359, we thank the reviewer for pointing out, "scale of" has been removed.

— line 328 : Maybe 2001 instead 2003 ?

Line 397, we thank the reviewer for correcting that, it is "2001" and has been corrected.